# Understanding the Transferability of Representations via Task-Relatedness

**Akshay Mehra, Yunbei Zhang, and Jihun Hamm**
Tulane University
{amehra, yzhang111, jhamm3}@tulane.edu

## Abstract

The growing popularity of transfer learning, due to the availability of models pre-trained on vast amounts of data, makes it imperative to understand when the knowledge of these pre-trained models can be transferred to obtain high-performing models on downstream target tasks. However, the exact conditions under which transfer learning succeeds in a cross-domain cross-task setting are still poorly understood. To bridge this gap, we propose a novel analysis that analyzes the transferability of the representations of pre-trained models to downstream tasks in terms of their relatedness to a given reference task. Our analysis leads to an upper bound on transferability in terms of task-relatedness, quantified using the difference between the class priors, label sets, and features of the two tasks. Our experiments using state-of-the-art pre-trained models show the effectiveness of task-relatedness in explaining transferability on various vision and language tasks. The efficient computability of task-relatedness even without labels of the target task and its high correlation with the model's accuracy after end-to-end fine-tuning on the target task makes it a useful metric for transferability estimation. Our empirical results of using task-relatedness on the problem of selecting the best pre-trained model from a model zoo for a target task highlight its utility for practical problems.

## 1   Introduction

Transfer learning (TL) [42, 59] is a powerful tool for developing high-performance machine learning models, especially in current times when large models [45, 11, 12, 18] pre-trained on huge amounts of data are being fine-tuned for various downstream tasks. While large pre-trained models achieve impressive performance on downstream tasks even in the zero-shot inference setting [45], their performance can often be improved by fine-tuning them on data from target tasks. However, our understanding of when representations from these models lead to classifiers that achieve high performance (i.e., high transferability) to downstream tasks is still lacking.

Analytical works based on domain adaptation [8, 7, 52, 36, 32, 37, 38] can only explain cross-domain tasks (i.e., when only features/label priors change across tasks) but in the TL setting, label sets can also change (i.e., cross-task setting). Recently, [56] showed that the relatedness between the label sets of the two tasks measured using conditional entropy can explain the difference in their transferability. However, [56] focused only on the cross-task setting, and analysis for transferability in a general cross-domain cross-task setting is not addressed. Apart from these analytical works, another line of work focuses on proposing transferability metrics that correlate well with performance on downstream tasks after end-to-end fine-tuning. We refer to these works as score-based transferability estimation (SbTE) metrics [61, 55, 29, 40, 19, 51]. These works focus on developing scores for selecting a pre-trained model from a model zoo, that achieves the best transferability on a target task. While these works address a practical problem, they do not focus on providing an analysis of transferability.

38th Conference on Neural Information Processing Systems (NeurIPS 2024).

Thus, we first rigorously analyze the transferability of the representations in producing high-performing classifiers and propose a novel approach that studies transferability in terms of its relatedness to a reference task (see Fig. 1). This is in line with previous analytical works [7, 2, 56, 55] which studied the model's performance on target tasks in terms of the source task in different settings such as domain adaptation/generalization and recently TL. However, there's a crucial difference: we study transferability in terms of a reference task instead of the source task since it is impractical to assume the knowledge of the source task used to train large models such as CLIP [45] or GPT, commonly used for TL.

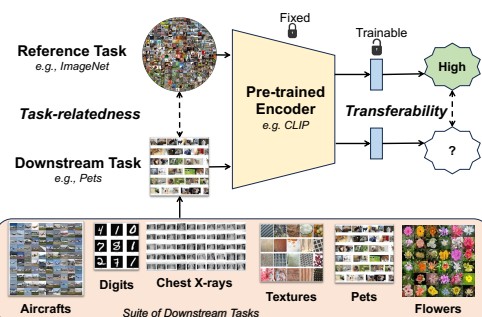

Figure 1: Given a pre-trained encoder (e.g., CLIP [45]), how does the performance after fine-tuning it on a reference task (e.g., ImageNet) relate to the performance after fine-tuning it on other tasks? Through a rigorous bound on transferability (Theorem 3) in terms of the relatedness between a reference and a target task, we show that tasks related to the reference task achieve provably better performance after fine-tuning.

Our approach works by transforming the distribution (and classifier) of a reference task, by transforming its class-prior distribution, label set, and feature space to obtain a new distribution that is similar to that of the target task (Fig. 2). Based on these transformations, we show that transferability can be provably explained (and is tightly upper bounded) using three interpretable terms. A weighted reference loss term appearing due to the class prior distribution difference between the tasks, a label mismatch term appearing as conditional entropy between the label distributions of the tasks, and a distribution mismatch term appearing as the Wasserstein distance between the transformed reference and target distributions (Theorem 3). We define task-relatedness as the sum of these three terms (a smaller value implies higher relatedness). We propose an optimization problem (Eq. 4) and an algorithm (Alg. 1) to learn the transformations to compute it. Using state-of-the-art (SOTA) pre-trained models, with different architectures, trained with various training methods on computer vision (CV) and natural language processing (NLP) tasks, we show that task-relatedness achieves a small gap to transferability (Sec. 4.1). Our analysis also leads to new insights into learning in the TL setting such as to improve the transferability of an encoder on a downstream task, one can improve the encoder's transferability on related reference tasks (Sec. 4.2). This is particularly useful when practitioners intend to develop encoders that achieve high transferability to proprietary (and potentially inaccessible) datasets.

We also demonstrate the utility of task-relatedness in estimating the accuracy of the model after end-to-end fine-tuning. While the TL setting assumes access to target labels, the high computational cost of end-to-end fine-tuning of a pre-trained model on a target task calls for developing metrics that are efficiently computable and highly correlated with end-to-end fine-tuning accuracy. To this end, we propose to use task-relatedness computed in the penultimate layer of the pre-trained model as our transferability estimation metric. To further improve the computational efficiency of task-relatedness we only measure the difference between the class-wise means and covariances of the distributions in lieu of the Wasserstein distance as required in Theorem 3. This enables the computation of task-relatedness with only the statistics of the reference/target tasks. Our empirical results (Sec. 4.3) attest that task-relatedness achieves a high correlation with the model's accuracy after end-to-end fine-tuning on the target task making it an effective metric for selecting a pre-trained model from a model zoo that achieves the best accuracy on the target task. Moreover, unlike previous SbTE metrics, task-relatedness can be estimated even without labeled target data, making it suitable for unsupervised transferability estimation, highlighting the advantage of a reference task as used in our analysis. Our main contributions are:

- We rigorously analyze transferability for classification tasks. Our analysis, to the best of our knowledge, leads to the first upper bound on transferability in terms of task-relatedness in a cross-domain cross-task setting.

- We propose an optimization problem to efficiently compute task-relatedness, using a small amount of target labels and show that it can even predict performance after end-to-end fine-tuning without requiring target labels.

- Using SOTA models and CV/NLP tasks, we show that task-relatedness accurately predicts transferability and show that transferability to unseen tasks can be improved by improving transferability to known (related) tasks.

## 2  Related Work

**Transfer learning (TL):** TL [42, 59, 18, 49, 46, 15, 14] has been studied widely and consists of various settings including transductive transfer, inductive transfer, and task transfer learning. The transductive setting also referred to as domain adaptation [8, 7] focuses on reducing the shift between two domains. The task transfer setting focuses on identifying the relationship between tasks, regardless of the model, to explain the transfer performance (see Appendix B for more details). Lastly, the inductive transfer setting focuses on using an inductive bias such as fine-tuning a pre-trained model (trained via adversarial training [48], self-supervised learning [11, 10, 12] or by combining language and image information [45]) to improve the performance on a target task. Our work focuses on the inductive transfer learning setting and proposes an upper bound on transferability of the representations of pre-trained models to downstream tasks.

**Analytical works for learning under distribution shifts:** Prior works [8, 7, 52, 36, 32, 37, 38] analytically explained learning under distribution shifts using distributional divergence between the marginal distributions and a label mismatch term. However, these results are applicable under assumptions such as covariate or label shift which need not be satisfied in TL where both the data distribution and the label spaces can be different (see App. B for detailed comparison). Recently, [56] proposed an upper bound on transferability in a restrictive setting of the same features for both tasks, however, our analysis does not require such an assumption. Other works [9, 47, 41] analyzed the representation for the multi-task learning setting. These works showed that when tasks are weakly related, a single representation space (model) may not perform well for all tasks. However, the TL setting differs from both of these and our work aims to analyze transferability in this setting.

**Score-based transferability estimation (SbTE):** These works [5, 40, 29, 61, 55, 39] use data from the target task and produce a score correlated with transferability. Such a score is useful for selecting the model from a model zoo that leads to the best transferability to a target task. [56] proposed the Negative Conditional Entropy (NCE) score that predicts transferability using the negative conditional entropy between labels of the tasks but requires the two tasks to have the same input instances. [6] estimates transferability by solving the HGR maximum correlation problem and using normalized Hscore, in the same setting as [56]. [40] proposed the LEEP score and computed NCE using soft (pseudo) labels for the target task from a pre-trained model. OT-CE [55] combined Wasserstein distance [3] and NCE whereas [5, 61] estimate likelihood and the marginalized likelihood of labeled target examples to estimate transferability. [33] proposes a model-agnostic approach that also relies on optimal transport to compute the distance between the tasks similar to OTDD [3]. In contrast, we focus on analyzing transferability in terms of task-relatedness theoretically along with demonstrating its effectiveness as a transferability estimation metric for the pre-trained model selection problem.

## 3  Analysis of TL using task-relatedness

**Problem setting and notations:** Let $P_R(x, y)$ and $P_T(x, y)$ denote the distributions of the reference and the target tasks, defined on $\mathcal{X}_R \times \mathcal{Y}_R$ and $\mathcal{X}_T \times \mathcal{Y}_T$ respectively. We assume that the feature spaces are common ($\mathcal{X}_R = \mathcal{X}_T = \mathcal{X}$) such as RGB images, but the reference label set $\mathcal{Y}_R = \{1, 2, \cdots, K_R\}$ and the target label set $\mathcal{Y}_T = \{1, 2, \cdots, K_T\}$ can be entirely different. We assume the number of reference task classes ($K_R$) are greater than or equal to the number of target classes ($K_T$). In the TL setting, an encoder (feature extractor) $g : \mathcal{X} \to \mathcal{Z}$ is pre-trained on a dataset with or without labels depending on the training method (e.g., supervised vs. self-supervised). We denote the resultant push-forward distributions of $R$ and $T$ on the encoder output space as $P_R(z, y)$ and $P_T(z, y)$. With a fixed encoder $g$, a classifier (linear or non-linear), $h(z) : \mathcal{Z} \to \Delta$, that outputs a probability vector is learned for the reference ($h_R$) and the target ($h_T$) separately, where $\Delta_{R/T}$ is a $K_R/K_T$ simplex for $R/T$. The classifier $h_R = \arg\min_{h \in \mathcal{H}} \mathbb{E}_{(z,y) \in P_R}[\ell(h(z; g), y)]$ and $h_T = \arg\min_{h \in \mathcal{H}} \mathbb{E}_{(z,y) \in P_T}[\ell(h(z; g), y)]$ where $\mathcal{H}$ is the set of classifiers and $\ell(h(z), y) = -\log(h(z)_y)$ is the cross-entropy loss. Table 3 in App. A summarizes the notations used in our work. Next, we define transferability as commonly used in the literature.

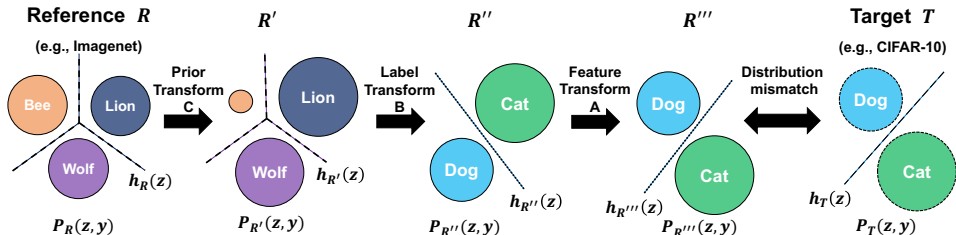

Figure 2: : **Overview of our task transformation model:** A series of transformations are applied to the reference distribution $P_R(z, y)$ and classifier $h_R$ to produce the transformed distribution $P_{R'''}$ and classifier $h_{R'''}$ to explain transferability to the downstream target task. Class-prior transformation $(R \to R')$ changes the class prior of the reference distribution (e.g., an irrelevant Bee class in $R$ now has smaller prior) followed by label set transformation $(R' \to R'')$ (e.g., to match {Lion, Wolf} with {Cat, Dog}), followed by feature space transformation $(R'' \to R''')$ to match the feature distribution of the target task $P_T(z, y)$.

**Definition 1.** (Transferability). Transferability of the representations from an encoder $g$ on a target task $T$ for classifiers in $\mathcal{H}$ is defined as $\mathbb{E}_{(z,y) \in P_T}[\ell(h_T(z; g), y)]$.

In the next section, we show the analysis with $\mathcal{H}$ as the class of linear classifiers for ease of explanation and discuss its extension to non-linear classifiers in App. A.5. Proofs for Sec. 3 are in App. A.

### 3.1 Our task transformation model

The reference and the target tasks share the same encoder but do not share label sets or data distributions. Therefore, to relate the two tasks, we propose a chain of three simple transformations: 1) prior transformation (from $R$ to $R'$), 2) label transformation (from $R'$ to $R''$), and 3) feature transformation (from $R''$ to $R'''$). The $R', R'', R'''$ are intermediate domain names after each of the transformations are applied. The corresponding classifier in each domain is denoted by $h_{R'}, h_{R''}$, and $h_{R'''}$ as illustrated in Fig. 2. The distribution after the transformations ($P_{R'''}$) has the same feature $\mathcal{Z}_{R'''} = \mathcal{Z}_T = \mathcal{Z}$ and label sets $\mathcal{Y}_{R'''} = \mathcal{Y}_T$ as the target task $T$, and consequently, the loss of the transformed classifier $h_{R'''}$ and the target classifier $h_T$ can be related.

**Class-prior transformation** $(R \to R')$**:** Since the reference task has more classes than the target task ($K_R \geq K_T$), many of the reference task classes are likely irrelevant for transfer to the target classes, e.g., while transferring from ImageNet to CIFAR10, only a small portion of ImageNet classes are relevant to CIFAR10 classes. The prior transformation accounts for the relative importance of the reference classes. This is illustrated in Fig. 2, where changing the class prior of $R$ reduces the prior of the Bee class and increases the priors of Wolf and Lion classes (shown by the changed size of classes Wolf and Lion in $R'$). While transforming the prior of $R$, we keep the conditional distribution and the classifier the same i.e., $P_{R'}(z|y) = P_R(z|y)$ and $h_{R'}(z) = h_R(z)$. Lemma 1 in App. A.2.1 shows that the expected loss of the classifier $h_R$ on $R'$ is a re-weighted version of the loss of $h_R$ on $R$.

**Label transformation** $(R' \to R'')$**:** Next, we use a label transformation to match the label sets of the new domain $R''$ and that of the target domain. To this end, we specify the conditional distribution $B_{ij} := P(y_{R''} = i|y_{R'} = j)$ ($B_{ij} \in [0, 1]$, $\forall i, j$, $\sum_i B_{ij} = 1$, $\forall j$). The label $y_{R''}$ of an example from the domain $R''$ is obtained via $BP(y_{R'})$. This generative process doesn't require the feature, i.e., $P_{R''}(y_{R''}|y_{R'}, z) = P_{R''}(y_{R''}|y_{R'})$. $B$ with sparse entries (i.e., only one entry of a column is 1) models a deterministic map from $\mathcal{Y}_R$ to $\mathcal{Y}_T$; $B$ with dense entries models a weaker association. This process is illustrated in Fig. 2 which shows the map from {Bee, Wolf, Lion} $\subset \mathcal{Y}_{R'}$ to {Dog, Cat} $\subset \mathcal{Y}_T$ after using $B$. Under this model, a reasonable choice of classifier for $R''$ is $h_{R''}(z) = Bh_{R'}(z)$. Lemma 2 in App. A.2.2 shows that the expected loss of $h_{R''}$ depends on the loss of $h_{R'}$ and the conditional entropy between the label sets of the tasks $R'$ and $R''$ and Corollary 1 shows the conditions for optimality of $h_{R''}$.

**Feature transformation** $(R'' \to R''')$**:** The final step involves changing the feature space of the distribution $R''$. We apply an invertible linear transformation $A$ to the distribution in $R''$ to obtain the new distribution $R'''$. After the transformation, the classifier associated with the new domain $R'''$ is $h_{R'''}(z) = h_{R''}(A^{-1}(z))$. This is illustrated in Fig. 2 after feature transform using $A$. Lemma 3 in

App. A.2.3 shows that a linear transform of the space and classifier does not incur any additional loss and Corollary 2 shows that the optimality of $h_{R''}$ implies optimality of $h_{R'''}$. Using these, we get Theorem 1 by defining conditional entropy as follows

$$H(\mathcal{Y}_{R''}|\mathcal{Y}_{R'}) = - \sum_{y_{R'} \in \mathcal{Y}_{R'}} \sum_{y_{R''} \in \mathcal{Y}_{R''}} P_{R'}(y_{R'}) B_{y_{R''},y_{R'}} \log(B_{y_{R''},y_{R'}}). \tag{1}$$

**Theorem 1.** *Let $C := \left[\frac{P_{R'}(y)}{P_R(y)}\right]_{y=1}^{K_R}$ be a vector of probability ratios , $B$ be a $K_T \times K_R$ matrix with $B_{ij} = P(y_{R''} = i|y_{R'} = j)$, $A : \mathcal{Z} \to \mathcal{Z}$ be an invertible linear map of features. Let the classifiers $h_{R'}(z) := h_R(z)$, $h_{R''}(z) := Bh_{R'}(z)$, $h_{R'''}(z) := h_{R''}(A^{-1}(z))$. Assuming $\ell$ is the cross-entropy loss, we have*

$$\mathbb{E}_{P_{R'''}(z,y)}[\ell(h_{R'''}(z),y)] \leq \underbrace{\mathbb{E}_{P_R(z,y)}[C(y)\ell(h_R(z),y)]}_{\textit{Re-weighted reference loss}} + \underbrace{H(\mathcal{Y}_{R''}|\mathcal{Y}_{R'})}_{\textit{Label mismatch}}.$$

Theorem 1 provides an upper bound on the loss of the final transformed classifier/distribution in terms of the loss of the reference classifier/distribution. The *re-weighted reference loss* shows that the performance of the transformed classifier on the new domain is linked to the label-wise re-weighted loss of the reference classifier on $R$. This implies that one can use only the relevant reference classes to contribute to the bound. The *label mismatch* term shows that the performance of the distribution $R'''$ and $R$ depends on the conditional entropy $H(\mathcal{Y}_{R''}|\mathcal{Y}_{R'}; B)$ between the label distributions of the domain $R''$ and $R'$. A high value of $H$ implies that the labels of the reference task are unrelated leading to lower transferability, whereas a low $H$ implies higher transferability. Corollary 3 in App. A.2.4 shows when the bound in Theorem 1 becomes equality.

## 3.2 Distribution mismatch between $P_{R'''}$ and $P_T$

After the three transformations, the transformed reference $P_{R'''}(z,y)$ can be compared with the target $P_T(z,y)$. However, these are only simple transformations and $P_{R'''}$ cannot be made identical to $P_T$ in general. This mismatch can be measured by the Wasserstein or Optimal Transport distance [44, 58]. Since our goal is to match two joint distributions defined on $\mathcal{Z} \times \mathcal{Y}$ we use

$$d((z,y),(z',y')) := \|z - z'\|_2 + \infty \cdot 1_{y \neq y'}, \tag{2}$$

with $z, z' \in \mathcal{Z}$ and $y, y' \in \mathcal{Y}$ as our base distance [53] to define the (type-1) Wasserstein distance

$$W_d(P,Q) := \inf_{\pi \in \Pi(P,Q)} \mathbb{E}_{((z,y),(z',y')) \sim \pi}[d((z,y),(z',y'))]. \tag{3}$$

Using Eq. 2, the Wasserstein distance between the joint distributions is the weighted sum of the Wasserstein distance between conditional distributions $(P(z|y))$ (Lemma 4 in App. A). Theorem 2 below explains the gap between the losses due to the distribution mismatch.

**Assumption 1.** 1) The composition of the loss function and the classifier $\ell \circ h$ is a $\tau-$Lipschitz function w.r.t to $\| \cdot \|_2$ norm, i.e., $|\ell(h(z),y) - \ell(h(z'),y)| \leq \tau\|z - z'\|_2$ for all $y \in \mathcal{Y}$, $z, z' \in \mathcal{Z}$ where $h \in \mathcal{H}$. 2) $P_T(y) = P_{R'''}(y)$.

The assumption 2), can be satisfied since we have full control on the prior $P_{R'''}(y)$ via $B$ and $C$.

**Theorem 2.** *Let the distributions $T$ and $R'''$ be defined on the same domain $\mathcal{Z} \times \mathcal{Y}$ and assumption 1 holds, then*

$$\mathbb{E}_{P_T(z,y)}[\ell(h(z),y)] - \mathbb{E}_{P_{R'''}(z,y)}[\ell(h(z),y)] \leq \underbrace{\tau\, W_d(P_{R'''},P_T)}_{\textit{Distribution mismatch}},$$

*with $d$ as in Eq. 2.*

Theorem 2 shows that when $\ell \circ h$ is $\tau-$Lipschitz then the performance gap between the $R'''$ and $T$ is bounded by the type-1 Wasserstein distance between the two distributions. The Lipschitz coefficient of the composition can be bounded by $\tau$, by penalizing the gradient norm w.r.t $z$ at training time. Thus, for linear fine-tuning, we train the classifiers $h_R$ and $h_T$ with an additional gradient norm penalty $\max\{0, \|\nabla_z\ell(h(z),y)\|_2 - \tau\}$ to make them conform to the Lipschitz assumption (see App. C.3). Note that constraining the Lipschitz constant restricts the hypothesis class. The trade-off between the Lipschitz constant and the performance of $h$ is empirically evaluated in App. C.3.1.

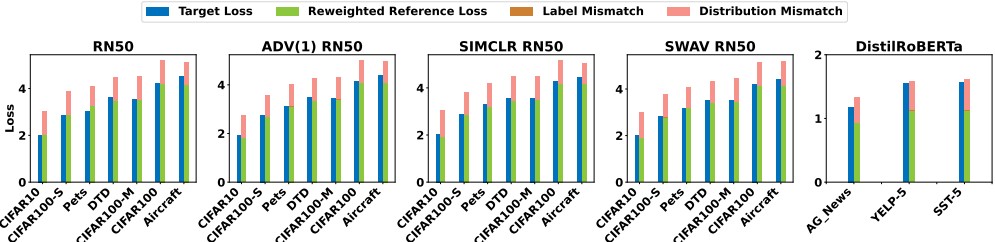

Figure 3: Task-relatedness (decomposed into its components) produces a small gap to transferability (blue bars). As the task-relatedness between the reference (ImageNet (for CV), DBPedia (for NLP)), and the target tasks (x-axis) increases, the transferability improves. (Note: the label mismatch term is zero in our figures as $B$ is fixed to a sparse matrix, see Sec. 3.4.)

### 3.3 Bounding transferability using task-relatedness

Here, we combine the results obtained in Theorem 1 and Theorem 2. The final bound proposed in Theorem 3 is one of our main contributions which explains transferability as a sum of three interpretable and measurable gaps.

**Theorem 3.** *Let $\ell$ be the cross-entropy loss, then under assumptions of Theorems 1 and 2,*
$$\mathbb{E}_{P_T(z,y)}[\ell(h_T(z),y)] \leq \underbrace{\mathbb{E}_{P_R(z,y)}[C(y)\ell(h_R(z),y)]}_{\text{Re-weighted reference loss}} + \underbrace{H(\mathcal{Y}_{R''}|\mathcal{Y}_{R'})}_{\text{Label mismatch}} + \underbrace{\tau\, W_d(P_{R'''}, P_T)}_{\text{Distribution mismatch}}.$$

The theorem shows that transferability can be decomposed into the loss incurred while transforming the class prior distribution, label space, and feature space of the reference distribution (first two terms) and the residual distance between the distribution obtained after transformations and the actual target distribution (last term). Based on the terms in the upper bound we define task-relatedness as follows.

**Definition 2.** (Task-relatedness). The relatedness between a target and a reference task is defined as $\mathbb{E}_{P_R(z,y)}[C(y)\ell(h_R(z),y)] + H(\mathcal{Y}_{R''}|\mathcal{Y}_{R'}) + \tau\, W_d(P_{R'''}, P_T)$.

A smaller value of the task-relatedness measure implies higher relatedness of the reference and the target tasks. In particular, when the target task is a transformation of the reference task then there exist transformations $A, B$, and $C$ such that the distribution $R'''$ perfectly matches the distribution of the target task (i.e., $W_d(P_{R'''}, P_T) = 0$). Moreover, when labels are deterministically related (Corollary 3) our bound becomes an equality.

Lastly, while we presented an analysis for linear fine-tuning here (for simplicity of presentation), our bounds hold for non-linear classifiers and non-linear feature transformations as well (see App. A.5).

### 3.4 Estimating task-relatedness

The optimization problem for learning the transformations $A, B$, and $C$ to compute task-relatedness in Theorem 3 is presented below. We use two new variables: inverse of the transformation $A$, denoted by $\bar{A} := A^{-1}$ and a transformed reference prior distribution denoted by $D(y) := C(y)P_R(y)$.

$$\min_{A,\bar{A},B,D} \mathbb{E}_{P_R(z,y)}\left[\frac{D(y)}{P_R(y)}\ell(h_R(z),y)\right] + H(\mathcal{Y}_{R''}|\mathcal{Y}_{R'}; B, D) + \tau W_d(P_{R'''}, P_T; A, B)$$

$$\text{s.t.} \quad A\bar{A} = \bar{A}A = I, \ \ P_T(y) = BD, \ \ \sum_i B_{ij} = 1 \ \forall j, \ \ \sum_{y \in \mathcal{Y}_R} D(y) = 1, \tag{4}$$

$$B_{ij} \in [0,1] \ \forall i,j, \ \text{and} \ D_i \in [0,1] \ \forall i.$$

Alg. 1 shows how we solve Eq. 4 (see App. D for additional details of the algorithm). Fig. 8 in App. C.1.2 shows how the upper bound is reduced as the optimization proceeds. Computationally, a single epoch of Alg. 1 takes a mere 0.17 seconds on our hardware for transfer from ImageNet to Pets for the ResNet-18 model (we ran Alg. 1 for 2000 epochs). In App. C.1, we show the effectiveness of learning the transformations using Alg. 1 on small-scale transfer tasks. Our results show that when

**Algorithm 1** Minimization of the bound in Theorem 3

---

**Input**: Reference task samples and labels $(Z_R, Y_R)$, Target task samples $(Z_T)$, Target task labels $(Y_T)$ (optional).
**Output**: Estimate of task-relatedness using the learned transformations $A, \bar{A}, B, D$.
**Init:** $A := \bar{A} := \mathbb{I}$, $D := P_R(y)$, random $B \in \mathbb{R}^{K_T \times K_R}$

1: Randomly sample $n_R$ points $(z_R^i, y_R^i) \sim (Z_R, Y_R)$ as per the class prior $D$.
2: **if** $Y_T$ is available **then**
3:     Randomly sample $n_T$ points $(z_T^j, y_T^j) \sim (Z_T, Y_T)$.
4: **else**
5:     Randomly sample $n_T$ points $(z_T^j) \sim (Z_T)$.
6:     # Compute pseudo-labels for the target samples $z_T$.
7:     $y_T^j = \arg\max_{y \in \mathcal{Y}_T} B h_R(A^{-1} z_T)$ for $j = 1, \cdots, n_T$.
8: **end if**
9: Compute $(z_{R'''}^i, y_{R'''}^i) = (A z_R^i, \ \arg\max_y B e(y_R^i))$, for $i = 1, \cdots, n_R$.
10: Assign $\mathcal{Y}_{R'} := \mathcal{Y}_R$ and $\mathcal{Y}_{R''} := \mathcal{Y}_T$.
11: Compute the optimal coupling $\pi^*$ between the distributions $R'''$ and $T$ by minimizing $W_d(P_{R'''}, P_T)$, i.e.,

$$\min_{\pi \in \Pi(P_{R'''}, P_T)} \quad \sum_{i,j} \pi_{ij} \tilde{d}((z_{R'''}^i, y_{R'''}^i), (z_T^j, y_T^j))$$

$$\text{s.t.} \quad \sum_j \pi_{ij} = \frac{1}{n_R} \ \forall i, \ \sum_i \pi_{ij} = \frac{1}{n_T} \ \forall j.$$

12: Using $\pi^*$, solve for $A, \bar{A}, B, D$ using mini-batch SGD

$$\min_{A, \bar{A}, B, D} \quad \sum_{i,j} \pi_{i,j}^* \left[ \tilde{d}((z_{R'''}^i, y_{R'''}^i), (z_T^j, y_T^j)) \right]$$

$$+ \ \frac{1}{n_R} \sum_i \frac{D(y^i)}{P_R(y^i)} \ell(h_R(z_R^i), y^i) + H(\mathcal{Y}_{R''} | \mathcal{Y}_{R'})$$

$$+ \ \|P_T(y) - BD\|_2^2 + (\|A\bar{A} - I\|_F + \|\bar{A}A - I\|_F).$$

13: Repeat 1 - 12 until convergence.

---

the reference task has classes semantically related to the target task, Alg. 1 learns transformations that achieve the smallest gap to transferability. However, since finding data semantically related to the target task may not always be possible we choose a reference task with the same number of classes as the target and fix that matrix $B$ to a random permutation of identity (making the label mismatch term zero) and $D$ to the prior of the reference task, learning only the transformation $A$, in our experiments.

## 4 Empirical Analysis

Here, we empirically demonstrate the effectiveness of task-relatedness in explaining transferability in various settings. We present additional results in App. C and dataset/experimental details in App. D. Our codes can be found at `https://github.com/akshaymehra24/TaskTransferAnalysis`.

### 4.1 Task-relatedness achieves a small gap to actual transferability

*Task-relatedness tightly upper bounds transferability across various architectures, pretraining methods, and datasets.* We demonstrate this by using various pre-trained models with architectures such as Vision Transformers (ViT) [20], ResNet-18/50/101/152 [27], DistilRoBERTa [34] trained with various pretraining methods including supervised training, adversarial training [48], SimCLR [11], MoCo [26], SwAV [10], and MAE [25]. We also consider a wide range of target datasets including, CIFAR10/100, Aircraft, Pets, DTD, AG-News, Yelp-5, and SST-5 whose details are in App. D.

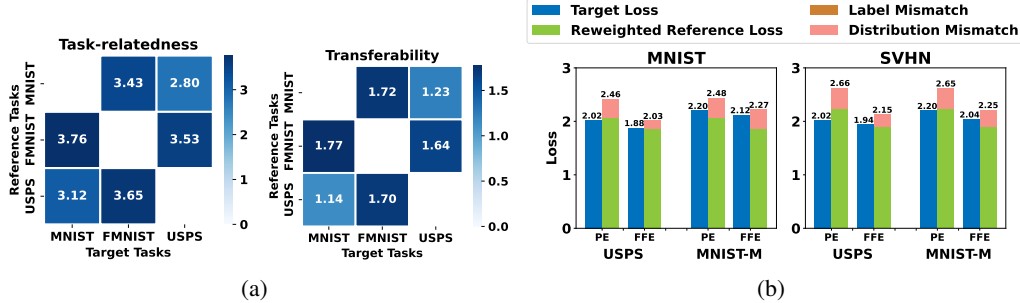

Figure 4: (a) Task-relatedness and transferability are highly correlated across various reference-target pairs. (b) Improving the transferability of an encoder on a reference task (in the plot title) leads to improved transferability of all related target tasks (x-axis). (e.g., compared to the original pre-trained CLIP encoder (PE), a end-to-end fine-tuned CLIP encoder (FFE) on the reference task achieves higher transferability to all related tasks.)

For this experiment, we fix the reference task to be ImageNet [17] for image classification and to DBPedia for sentence classification tasks and use Alg. 1 to estimate task-relatedness. The results in Fig. 3 and 9 (in the Appendix) show that our bound achieves a small gap to actual transferability. As the task-relatedness between the reference and the target tasks improves, transferability also improves showing that task-relatedness and transferability are strongly correlated. *Task-relatedness is also strongly correlated with the accuracy of the end-to-end fine-tuned classifiers on the target task.* In Fig. 11 (in the Appendix), we show high Pearson correlation coefficients ($\geq -0.57$) for task-relatedness and accuracy after fully fine-tuning various pre-trained encoders using data from various target tasks.

## 4.2 Effect of the reference task on task-relatedness

*Highly related reference–target task pairs, based on task-relatedness, achieve higher transferability coinciding with the semantic relatedness between tasks.* To understand how a reference task affects task-relatedness and eventually transferability, we consider two experiments using convolutional and CLIP-trained models with various character recognition tasks such as MNIST, Fashion-MNIST (FMNIST), SVHN, MNIST-M, and USPS. Of these datasets, SVHN and MNIST-M contain colored images while the rest contain gray-scale images. In the first experiment, we train convolutional models on MNIST, FMNIST, and USPS and measure pairwise transferability. Here we use the reference task to be the same task as that used for training the models. The results in Fig. 4(a) show that transferability to those target tasks is higher for which task-relatedness metric's value is smaller. Specifically, USPS achieves the best transferability (1.23) and the smallest task-relatedness (2.80) when the reference task is MNIST. This is attributed to both datasets containing gray-scale images of digits. On the other hand, when the reference task is unrelated to the target task i.e., the task-relatedness value is high, transferability suffers, e.g., when the reference task is MNIST and the target task is FMNIST. Results in App. C.2.2 show similar results for the sentence classification task.

*The gap between task-relatedness and transferability is smaller when a reference task performs well with a given encoder.* Here we use MNIST and SVHN as two reference tasks and compute the task-relatedness and transferability with USPS and MNIST-M as target tasks, using CLIP (Vit B32) model. A linear classifier trained on top of the embeddings from the CLIP model achieves ≈98% accuracy on MNIST but only ≈61% accuracy for SVHN. Due to this, transferability (USPS:2.02, MNIST-M:2.20) explained using task-relatedness with MNIST as the reference task (USPS:**2.46**, MNIST-M:**2.48**) is better than that computed using SVHN (USPS:2.66, MNIST-M:2.65) as the reference, even though MNIST-M is intuitively more similar to SVHN (as both contain colored images of digits). This is evident from the results of PE (Pre-trained Encoder) in Fig. 4(b).

*Improving the performance of an encoder on a reference task improves transferability to other related (potentially unseen) tasks.* To show this we fully fine-tune the CLIP encoder on MNIST and SVHN tasks, increasing the accuracy of the classifiers for both MNIST and SVHN to 99% and 95%, respectively. Using the representations from these new encoders, we find that the transferability of both related target tasks improves along with task-relatedness (see FFE results in Fig. 4(b)). Here,

Table 1: Task-relatedness achieves high (negative) Pearson correlation to the accuracy after end-to-end fine-tuning for various tasks. For NCE [56], Leep [40], LogMe [61], SFDA [51], OT-NCE, OTCE [55], and H-score [6] **positive** correlation is better whereas for PACTran [19] and task-relatedness (ours) **negative** correlation is better.

| Target task | LogMe | Leep | NCE | PACTran | SFDA | H-Score | OT-NCE | OTCE | Ours |
|---|---|---|---|---|---|---|---|---|---|
| Pets | 0.82 | 0.80 | 0.73 | -0.82 | 0.57 | 0.77 | 0.88 | 0.86 | -0.77 |
| DTD | 0.88 | 0.96 | -0.19 | -0.85 | 0.90 | 0.89 | 0.84 | 0.82 | -0.97 |
| Aircraft | -0.60 | 0.92 | 0.97 | 0.11 | 0.72 | -0.80 | 0.56 | 0.60 | -0.72 |
| **Average** | 0.37 | 0.90 | 0.50 | -0.52 | 0.73 | 0.29 | 0.76 | 0.76 | -0.82 |

we see that task-relatedness for MNIST-M and USPS is the best when the reference task is SVHN and MNIST, respectively, aligning with our intuition of semantic relatedness between these tasks. This also suggests that transferability on other related tasks can be improved by fully fine-tuning the encoder on these reference tasks. Thus, in scenarios where target tasks are private (such as proprietary Chest X-rays), an encoder trained to work well on related tasks (such as publicly available Chest X-rays) is bound to achieve good transferability.

### 4.3 Task-relatedness for end-to-end transferability estimation

In this section, we show an efficient way of computing task-relatedness which enables its use for estimating transferability after end-to-end fine-tuning. While Alg. 1, accurately estimates task-relatedness by minimizing the bound in Eq. 3, it could be inefficient due to the requirement of computing and minimizing the Wasserstein distance between distributions at every epoch. Thus, to make the computation efficient, we replace the Wasserstein distance computation in step 11 and 12 of Alg. 1, with mean and covariance matching terms. Specifically, we define the distance between two distributions $R'''$ and $T$ as

$$\Gamma(R''', T) := \|\mu_{R'''} - \mu_T\|_2^2 + \lambda \|\Sigma_{R'''} - \Sigma_T\|_2^2, \tag{5}$$

where $\mu_{R'''/T} := \frac{1}{n_{R'''/T}} \sum_{z \in P_{R'''/T}} z$, $\Sigma_{R'''/T} := \frac{1}{n_{R'''/T}} \sum_{z \in P_{R'''/T}} (z - \mu_{R'''/T})^T (z - \mu_{R'''/T})$, and $\lambda$ is a regularization coefficient. Using $\Gamma(R''', T)$ in place of $W_d(R''', T)$, makes the computation of task-relatedness by learning transformations $A$, $B$, and $C$ significantly more efficient.

*Task-relatedness is an effective metric for the pre-trained model selection problem.* The goal of this problem is to find a pre-trained model from a model zoo that achieves the best accuracy on a given target task after end-to-end fine-tuning of the model using labeled target data. Since end-to-end fine-tuning is costly (takes almost a day to fully fine-tune a single model on a single target task as shown by [61]), an effective transferability metric is significantly more efficient to compute and is correlated well with the accuracy after end-to-end finetuning. Using 5 different pre-trained models (supervised ResNet-50/101/152, adversarially pre-trained [48] ResNet50 with $\epsilon \in \{0.1, 1\}$) and ImageNet as the reference task, we show in Table 1, that task-relatedness achieves a high correlation with the accuracy after end-to-end fine-tuning on the target task. Our results also highlight the instability of various popular SbTE metrics, such as LogMe [61] and NCE [56] which can produce a high negative correlation, and those of PACTran [19] which achieve low correlation values on complex datasets. In comparison, task-relatedness consistently achieves a good correlation for various target tasks. Computationally, it takes a mere 3-4 minutes to learn the transformations to compute task-relatedness, providing a significant computation advantage over end-to-end fine-tuning. We also show that *task-relatedness remains highly correlated with end-to-end fine-tuning accuracy even with a limited amount of labeled data from the target task* as shown in Fig. 5 unlike other SbTE metrics.

Next, we show that *task-relatedness can even be estimated without using labels from the target task.* For scenarios, where labeled data from the target task is unavailable, estimating transferability is challenging. This is because both fine-tuning and most SbTE methods require labels to compute the transferability scores. Here we show that task-relatedness can still be an effective measure to estimate transferability in this challenging setting. Since we use a transformative model and have access to a reference task/classifier, we can use the predictions from the reference task's classifier trans-

Table 2: Correlation of task-relatedness and end-to-end fine-tuning accuracy computed using true and pseudo labels of the target task.

| Target | True labels | Pseudo labels |
|---|---|---|
| Pets | -0.77 | -0.76 |
| DTD | -0.97 | -0.91 |
| Aircraft | -0.72 | -0.16 |

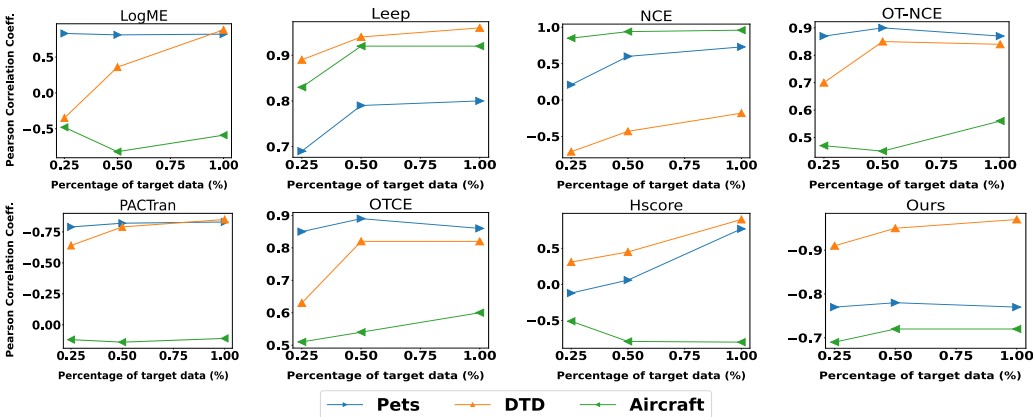

Figure 5: Task-relatedness (Ours) remains highly correlated with accuracy after end-to-end fine-tuning on a target task even when using a small percentage of target data unlike other SbTE methods (LogME, Leep, NCE, PACTran, OT-NCE, OTCE, and H-Score) whose correlation is affected significantly. For LogMe, Leep, NCE, OT-NCE, OTCE, and H-score **positive** correlation is better whereas for PACTran and task-relatedness (ours) **negative** correlation is better.

formed via $B$ (to obtain labels $\in \mathcal{Y}_T$) and estimate the *pseudo*-labels of the target data. Concretely, pseudo-label for a target sample $x_T$ is obtained as $y_T^{pseudo} = \arg\max_{y \in \mathcal{Y}_T} B h_R(A^{-1}(z_T))$. Results in Table 2 show that our task-relatedness estimated via pseudo-labeled target data still achieves a high correlation to transferability on most datasets. For datasets such as Pets and DTD, where transforming the reference task classifier produces high accuracy on the target task, the difference between the pseudo and true labels is small. Consequently, the difference in the correlations with pseudo and true labels is also small. Thus, when the reference and target tasks are related, transferability can be estimated accurately without requiring labels of the target task, showing that task-relatedness is an effective metric even for unsupervised transferability estimation.

# 5 Conclusion

We analyzed TL in terms of the relatedness between the target and a reference task. Our analysis works by transforming the distribution of a reference task to match that of the target. Using this we proved an upper bound on transferability, defined as task-relatedness, consisting of three interpretable terms, namely, the re-weighted reference task loss, label mismatch, and distribution mismatch. We proposed an algorithm to compute task-relatedness and demonstrated its effectiveness at accurately predicting transferability (even without target labels) with SOTA models. Moreover, the high correlation of task-relatedness with accuracy after end-to-end fine-tuning and its efficient computability, makes it an effective metric for transferability estimation.

**Limitations.** We studied transferability using the cross-entropy loss and used Wasserstein distance-based distribution shift analysis due to their popularity. However, due to accuracy being the primary metric of interest in classification tasks and the difficulty of computing the Wasserstein distance with limited samples in a high dimensional representation space, extending the analysis to 0-1 loss and other divergence measures are important directions which are not addressed here and are left for future works.

# 6 Acknowledgment

We thank the anonymous reviewers of this work for their insightful comments and suggestions. This work was supported by the NSF EPSCoR-Louisiana Materials Design Alliance (LAMDA) program #OIA-1946231.

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

# Appendix

We present the missing proofs of the theoretical results from Sec. 3 along with justifications for the classifiers $(h_{R'}, h_{R''}, h_{R'''})$ as Corollaries in Appendix A followed by related work on learning in the presence of distribution shift with the same feature and label space in Appendix B. This is followed by additional experimental results including NLP classification tasks with large pre-trained models in Appendix C. We conclude in Appendix D with details of the experiments and datasets used.

## A  Proofs for Sec. 3

### A.1  Notation

Table 3: Table of notations

| | Data related |
|---|---|
| $R/T$ | Reference/Target task. |
| $\mathcal{X}_{R/T}$ | Images of reference/target tasks. |
| $\mathcal{Y}_{R/T}$ | Label set of reference/target tasks. |
| $K_{R/T}$ | Number of classes in reference/target tasks. |
| $P_{R/T}(x, y)$ | Data distribution of reference/target tasks. |
| $(Z_{R/T}, Y_{R/T})$ | Samples (features and labels) of the reference/target tasks. |
| | Model related |
| $g$ | Encoder pre-trained on a pre-training dataset. |
| $\mathcal{Z}_{R/T}$ | Representations extracted for reference/target tasks using $g$. |
| $h_{R/T}$ | Classifier learned for reference/target tasks on the representations of $g$. |
| $\ell$ | Cross-entropy loss. |
| $\tau$ | Lipschitz constant. |
| | Task-transformation related |
| $A$ | Parameters for feature transformation. |
| $B$ | Parameters for label transformation. |
| $C$ | Parameters for class-prior transformation. |
| $P_{R'}(z, y)$ and $h_{R'}$ | Distribution and classifier of $R'$ after applying transformation $C$ on $R$. |
| $P_{R''}(z, y)$ and $h_{R''}$ | Distribution and classifier of $R''$ after applying transformation $B$ on $R'$. |
| $P_{R'''}(z, y)$ and $h_{R'''}$ | Distribution and classifier of $R'''$ after applying transformation $A$ on $R''$. |
| $H$ | Conditional entropy as defined in Eq. 1. |
| $d$ | Base distance between two samples as defined in Eq. 2. |
| $W_d$ | Wasserstein distance between two distributions defined in Eq. 3. |
| $\Gamma$ | Distance between two distributions based on their mean/variance defined in Eq. 5. |

### A.2  Our task transformation model (Sec. 3.1)

#### A.2.1  Class-Prior transformation $(R \to R')$

**Lemma 1.** *Let $C := \left[\frac{P_{R'}(y)}{P_R(y)}\right]_{y=1}^{K_R}$ be a vector of probability ratios and the classifier $h_{R'}(z) := h_R(z)$, then $\mathbb{E}_{P_{R'}(z,y)}[\ell(h_{R'}(z), y)] = \mathbb{E}_{P_R(z,y)}[C(y)\ell(h_R(z), y)]$, for any loss function $\ell$.*

*Proof.*

$$\mathbb{E}_{P_{R'}(z,y)}[\ell(h_{R'}(z),y)] = \mathbb{E}_{P_{R'}(z,y)}[\ell(h_R(z),y)] = \sum_{y\in\mathcal{Y}_R} P_{R'}(y)\mathbb{E}_{P_{R'}(z|y)}[\ell(h_R(z),y)]$$

$$= \sum_{y\in\mathcal{Y}_R} \frac{P_R(y)}{P_R(y)} P_{R'}(y)\mathbb{E}_{P_{R'}(z|y)}[\ell(h_R(z),y)] = \sum_{y\in\mathcal{Y}_R} P_R(y)\mathbb{E}_{P_{R'}(z|y)}[C(y)\ell(h_R(z),y)]$$

$$= \sum_{y\in\mathcal{Y}_R} P_R(y)\mathbb{E}_{P_R(z|y)}[C(y)\ell(h_R(z),y)] \quad \text{(since } P_R(z|y) = P_{R'}(z|y) \text{ by construction)}$$

$$= \mathbb{E}_{P_R(z,y)}[C(y)\ell(h_R(z),y)].$$

$\square$

### A.2.2 Label transformation $(R' \rightarrow R'')$

**Lemma 2.** *Let $B$ be a $K_T \times K_R$ matrix with $B_{ij} = P(y_{R''} = i|y_{R'} = j)$ and $h_{R''}(z) := Bh_{R'}(z)$ and $\ell$ be the cross-entropy loss. Then, $\mathbb{E}_{P_{R''}(z,y)}[\ell(h_{R''}(z),y)] \leq \mathbb{E}_{P_{R'}(z,y)}[\ell(h_{R'}(z),y)] + H(\mathcal{Y}_{R''}|\mathcal{Y}_{R'})$, where $H(\mathcal{Y}_{R''}|\mathcal{Y}_{R'})$ is the conditional entropy $(-\sum_{y_{R'}\in\mathcal{Y}_{R'}} \sum_{y_{R''}\in\mathcal{Y}_{R''}} P_{R'}(y_{R'})B_{y_{R''},y_{R'}} \log(B_{y_{R''},y_{R'}}))$.*

*Proof.* Note that $P(z) := P_{R'}(z) = P_{R''}(z)$ by construction.

$$\mathbb{E}_{P_{R''}(z,y)}[\ell(h_{R''}(z),y)] = \mathbb{E}_{P(z,y'')}[\ell(h_{R''}(z),y'')]$$

$$= \mathbb{E}_{P(z)}\mathbb{E}_{P(y''|z)}[\ell(h_{R''}(z),y'')] = \mathbb{E}_{P(z)}\sum_{y''}\sum_{y'} P(y'',y'|z)(\ell(h_{R''}(z),y'')) \quad \text{(since } y' \in \mathcal{Y}_{R'})$$

$$= \mathbb{E}_{P(z)}\mathbb{E}_{P(y'',y'|z)}[\ell(h_{R''}(z),y'')]$$

$$= \mathbb{E}_{P(z)}\mathbb{E}_{P(y'|z)}\mathbb{E}_{P(y''|y')}[\ell(h_{R''}(z),y'')] \quad \text{(since } P(y''|y',z) = P(y''|y'))$$

$$= \mathbb{E}_{P(z)}\mathbb{E}_{P(y'|z)}[\sum_{y''\in\mathcal{Y}_{R''}} \ell(h_{R''}(z),y'')B_{y'',y'}] \quad \text{(since } B_{y'',y'} = P(y''|y'))$$

$$= \mathbb{E}_{P(z,y')}[\sum_{y''\in\mathcal{Y}_{R''}} \ell(Bh_{R'}(z),y'')B_{y'',y'}].$$

Since the loss $\ell$ is the cross-entropy loss, we have

$$\ell(Bh_{R'}(z),y'') = -\log(\sum_{j\in\mathcal{Y}_{R'}} B_{y'',j}h_{R'}^j(z)) \leq -\log(B_{y'',y'}h_{R'}^{y'}(z)) = -\log(B_{y'',y'}) - \log(h_{R'}^{y'}(z)).$$

Therefore, we have

$$\mathbb{E}_{P_{R''}(z,y)}[\ell(h_{R''}(z),y)]$$

$$= \mathbb{E}_{P(z,y')}[\sum_{y''\in\mathcal{Y}_{R''}}\ell(Bh_{R'}(z),y'')B_{y'',y'}]$$

$$\leq -\mathbb{E}_{P(z,y')}[\sum_{y''\in\mathcal{Y}_{R''}}B_{y'',y'}\left(\log(B_{y'',y'})+\log(h_{R'}^{y'}(z))\right)]$$

$$= -\mathbb{E}_{P(z,y')}[\sum_{y''\in\mathcal{Y}_{R''}}B_{y'',y'}\log(B_{y'',y'})]-\mathbb{E}_{P(z,y')}[\sum_{y''\in\mathcal{Y}_{R''}}B_{y'',y'}\log(h_{R'}^{y'}(z))]$$

$$= -\mathbb{E}_{P(z,y')}[\sum_{y''\in\mathcal{Y}_{R''}}B_{y'',y'}\log(B_{y'',y'})]+\mathbb{E}_{P(z,y')}[-\log(h_{R'}^{y'}(z))\sum_{y''\in\mathcal{Y}_{R''}}B_{y'',y'}]$$

$$= -\mathbb{E}_{P(z,y')}[\sum_{y''\in\mathcal{Y}_{R''}}B_{y'',y'}\log(B_{y'',y'})]+\mathbb{E}_{P(z,y')}[-\log(h_{R'}^{y'}(z))]$$

$$= \mathbb{E}_{P(z,y')}[-\sum_{y''\in\mathcal{Y}_{R''}}B_{y'',y'}\log(B_{y'',y'})]+\mathbb{E}_{P(z,y')}[\ell(h_{R'}(z),y')]$$

$$= \mathbb{E}_{P(y')}\mathbb{E}_{P_{R'}(z|y')}[-\sum_{y''\in\mathcal{Y}_{R''}}B_{y'',y'}\log(B_{y'',y'})]+\mathbb{E}_{P(z,y')}[\ell(h_{R'}(z),y')]$$

$$= [-\sum_{y'\in\mathcal{Y}_{R'}}\sum_{y''\in\mathcal{Y}_{R''}}P(y')B_{y'',y'}\log(B_{y'',y'})]+\mathbb{E}_{P(z,y')}[\ell(h_{R'}(z),y')]$$

$$= H(\mathcal{Y}_{R''}|\mathcal{Y}_{R'})+\mathbb{E}_{P_{R'}(z,y)}[\ell(h_{R'}(z),y)].$$

$\square$

Corollary 1 below, shows the conditions under which the optimal softmax classifier for the domain $R'$ remains optimal for the domain $R''$, justifying our choice of classifier change from $R'$ to $R''$.

**Corollary 1.** *Let $e$ be one-hot encoding of the labels, $|\mathcal{Y}_{R''}| = |\mathcal{Y}_{R'}|$, $B$ be a $K_T \times K_R$ permutation matrix and $h_{R'}$ be the optimal softmax classifier for $R'$ and $y_{R''} := \sigma(y_{R'}) := \arg\max_{y\in\mathcal{Y}_{R''}}(Be(y_{R'}))_y$ then under the assumptions of Lemma 2, $h_{R''}(z) := Bh_{R'}(z)$ is the optimal softmax classifier for $R''$.*

*Proof.* Since $y_{R''} := \sigma(y_{R'}) := \arg\max_{y\in\mathcal{Y}_T}(Be(y_{R'}))_y$ we have

$$\mathbb{E}_{P_{R''}}[\ell(h_{R''}(z),y_{R''})] = \mathbb{E}_{P(z,y'')}[\ell(Bh_{R'}(z),y'')] = \sum_{y''\in\mathcal{Y}_{R''}}P(y'')\mathbb{E}_{P(z|y'')}[\ell(Bh_{R'}(z),y'')]$$

$$= \sum_{y'\in\mathcal{Y}_{R'}}P(\sigma(y'))\mathbb{E}_{P(z|\sigma(y'))}[\ell(Bh_{R'}(z),\sigma(y'))]$$

$$= \sum_{y'\in\mathcal{Y}_{R'}}P(y')\mathbb{E}_{P(z|y')}[\ell(h_{R'}(z),y')] = \mathbb{E}_{P_{R'}}[\ell(h_{R'}(z),y_{R'})].$$

The second last equality follows due to the symmetry of cross-entropy loss, i.e., $\ell(h,y) = -\log h_y = -\log Bh_{\sigma(y)} = \ell(Bh,\sigma(y))$.

Since $\min_{h_{R''}}\mathbb{E}_{P_{R''}}[\ell(h_{R''}(z),y_{R''})] = \min_{h_{R'}}\mathbb{E}_{P_{R'}}[\ell(h_{R'}(z),y_{R'})]$ and $h_{R'}$ is optimal for $R'$ we have $h_{R''}(z) := Bh_{R'}(z)$ is the optimal softmax classifier for $R''$. $\square$

### A.2.3 Feature transformation $(R'' \to R''')$

**Lemma 3.** *Let $A : \mathcal{Z} \to \mathcal{Z}$ be an invertible linear map of features and the classifier $h_{R'''}(z) := h_{R''}(A^{-1}(z))$. Then $\mathbb{E}_{P_{R'''}(z,y)}[\ell(h_{R'''}(z),y)] = \mathbb{E}_{P_{R''}(z,y)}[\ell(h_{R''}(z),y)]$ for any loss $\ell$.*

*Proof.* $\mathbb{E}_{P_{R'''}(z,y)}[\ell(h_{R'''}(z),y)] = \mathbb{E}_{P_{R'''}(z,y)}[\ell(h_{R''}(A^{-1}(z)),y)] = \mathbb{E}_{P_{R''}(z,y)}[\ell(h_{R''}(z),y)].$ $\square$

Our Corollary 2 below shows that the optimal softmax classifier for domain $R''$ remains optimal for domain $R'''$ too.

**Corollary 2.** *Let $h_{R''}$ be the optimal softmax classifier in domain $R''$ then under the assumptions of Lemma 3, $h_{R'''}(z) = h_{R''}(A^{-1}(z))$ is the optimal softmax classifier in domain $R'''$.*

*Proof.* When $h_{R'''}(z) = h_{R''}(A^{-1}(z))$, $\min_{h_{R''}} \mathbb{E}_{P_{R''}(z,y)}[\ell(h_{R''}(z), y)] = \min_{h_{R'''}} \mathbb{E}_{P_{R'''}(z,y)}[\ell(h_{R'''}(z), y)]$ by Lemma 3, hence if $h_{R''}$ is optimal for $R''$ then so is $h_{R'''}$ for the domain $R'''$. $\qquad\square$

### A.2.4  Three transformations combined $(R \to R''')$

**Theorem 1.** *Let $C := \left[ \frac{P_{R'}(y)}{P_R(y)} \right]_{y=1}^{K_R}$ be a vector of probability ratios , $B$ be a $K_T \times K_R$ matrix with $B_{ij} = P(y_{R''} = i | y_{R'} = j)$, $A : \mathcal{Z} \to \mathcal{Z}$ be an invertible linear map of features. Let the classifiers $h_{R'}(z) := h_R(z)$, $h_{R''}(z) := Bh_{R'}(z)$, $h_{R'''}(z) := h_{R''}(A^{-1}(z))$. Assuming $\ell$ is the cross-entropy loss, we have*

$$\mathbb{E}_{P_{R'''}(z,y)}[\ell(h_{R'''}(z), y)] \leq \underbrace{\mathbb{E}_{P_R(z,y)}[C(y)\ell(h_R(z), y)]}_{\text{Re-weighted reference task loss}} + \underbrace{H(\mathcal{Y}_{R''} | \mathcal{Y}_{R'})}_{\text{Label mismatch}}.$$

*Proof.*

$$
\begin{aligned}
\mathbb{E}_{P_{R'''}(z,y)}[\ell(h_{R'''}(z), y)] &= \mathbb{E}_{P_{R''}(z,y)}[\ell(h_{R''}(z), y)] \text{ (Lemma 3)} \\
&\leq \mathbb{E}_{P_{R'}(z,y)}[\ell(h_{R'}(z), y)] + H(\mathcal{Y}_{R''} | \mathcal{Y}_{R'}) \text{ (Lemma 2)} \\
&= \mathbb{E}_{P_R(z,y)}[C(y)\ell(h_R(z), y)] + H(\mathcal{Y}_{R''} | \mathcal{Y}_{R'}) \text{ (Lemma 1)}.
\end{aligned}
$$

$\qquad\square$

**Corollary 3.** *Let $e$ be one-hot encoding of the labels, $|Y_{R'''}| = |Y_R|$, $B : \Delta_{R'} \to \Delta_{R''}$ be a permutation matrix and $y_{R''} := \sigma(y_{R'}) := \arg\max_{y \in \mathcal{Y}_{R''}} (Be(y_{R'}))_y$ then under the assumptions of Lemmas 1, 2, and 3 we have $\mathbb{E}_{P_{R'''}(z,y)}[\ell(h_{R'''}(z), y)] = \mathbb{E}_{P_R(z,y)}[C(y)\ell(h_R(z), y)]$.*

*Proof.* Since $y_{R''} := \sigma(y_{R'}) := \arg\max_{y \in \mathcal{Y}_T} (Be(y_{R'}))_y$ we have

$$
\begin{aligned}
\mathbb{E}_{P_{R''}}[\ell(h_{R''}(z), y_{R''})] &= \mathbb{E}_{P(z,y'')}[\ell(Bh_{R'}(z), y'')] = \sum_{y'' \in \mathcal{Y}_{R''}} P(y'') \mathbb{E}_{P(z|y'')}[\ell(Bh_{R'}(z), y'')] \\
&= \sum_{y' \in \mathcal{Y}_{R'}} P(\sigma(y')) \mathbb{E}_{P(z|\sigma(y'))}[\ell(Bh_{R'}(z), \sigma(y'))] \\
&= \sum_{y' \in \mathcal{Y}_{R'}} P(y') \mathbb{E}_{P(z|y')}[\ell(h_{R'}(z), y')] = \mathbb{E}_{P_{R'}(z,y)}[\ell(h_{R'}(z), y)].
\end{aligned}
$$

The second last equality follows due to the symmetry of cross-entropy loss, i.e., $\ell(h, y) = -\log h_y = -\log Bh_{\sigma(y)} = \ell(Bh, \sigma(y))$.

Therefore, we have

$$
\begin{aligned}
\mathbb{E}_{P_{R'''}(z,y)}[\ell(h_{R'''}(z), y)] &= \mathbb{E}_{P_{R''}(z,y)}[\ell(h_{R''}(z), y)] \text{ (Lemma 3)} \\
&= \mathbb{E}_{P_{R'}(z,y)}[\ell(h_{R'}(z), y)] \text{ (from above)} \\
&= \mathbb{E}_{P_R(z,y)}[C(y)\ell(h_R(z), y)] \text{ (Lemma 1)}.
\end{aligned}
$$

$\qquad\square$

### A.3  Distribution mismatch between $R'''$ and $T$ (Sec. 3.2)

**Lemma 4.** *Let $U$ and $Q$ be two distributions on $\mathcal{Z} \times \mathcal{Y}$ with the same prior $P_U(y = i) = P_Q(y = i) = P(y = i)$. With the base distance $d$ defined as in Eq. 2, we have $W_d(P_U, P_Q) = \sum_y P(y) W_{\|\cdot\|_2}(P_U(z|y), P_Q(z|y))$.*

*Proof.* Let $\omega_y^*$ denote the optimal coupling for the conditional distributions $(P_U(z|y), P_Q(z|y))$ for $y \in \mathcal{Y}$ and $\pi^*$ denote the the optimal coupling for the joint distributions $(P_U(z, y), P_Q(z, y))$. Then, under the definition of our base distance $d$, $\pi^*((z, y), (z', y')) = 0$ when $y \neq y'$ i.e. no mass from the distribution $U$ belonging to class $y$ can be moved to the classes $y' \neq y$ of the distribution $Q$ when the class priors of $U$ and $Q$ are the same. Moreover, since $\sum_{ij}(\omega_y^*)_{ij} = 1$ and $\sum_{\{i,j:y_i=y_j'=k\}} \pi_{ij}^* = P(y = k)$ for $k \in \mathcal{Y}$ we have $\pi^*((z, y), (z', y')) = \omega_y^*(z, z')P(y)1_{y=y'}$ for every $y, y' \in \mathcal{Y}$.

Then, we can show that the total Wasserstein distance between the joint distributions can be expressed as the sum of conditional Wasserstein distances, as follows

$$
\begin{aligned}
& W_d(P_U(z, y), P_Q(z, y)) \\
= \; & \sum_{y,y'} \int \pi^*((z, y), (z', y'))d((z, y), (z', y'))dzdz' \\
= \; & \sum_{y,y'} \int \pi^*((z, y), (z', y'))(\|z - z'\|_2 + \infty \cdot 1_{y \neq y'})dzdz' \\
= \; & \sum_{y,y'} \int \omega_y^*(z, z')P(y)1_{y=y'}(\|z - z'\|_2 + \infty \cdot 1_{y \neq y'})dzdz' \\
= \; & \sum_{y,y'} \int \omega_y^*(z, z')P(y)1_{y=y'}\|z - z'\|_2 dzdz' \quad (\text{since } 1_{y=y'} \cdot 1_{y \neq y'} = 0) \\
= \; & \sum_{y,y'} P(y)1_{y=y'} \int \omega_y^*(z, z')\|z - z'\|_2 dzdz' \\
= \; & \sum_{y} P(y) \int \omega_y^*(z, z')\|z - z'\|_2 dzdz' \\
= \; & \sum_{y} P(y)W_{\|\cdot\|_2}(P_U(z|y), P_Q(z|y)).
\end{aligned}
$$

$\square$

**Theorem 2.** *Let the distributions $T$ and $R'''$ be defined on the same domain $\mathcal{Z} \times \mathcal{Y}$ and assumption 1 holds, then $\mathbb{E}_{P_T(z,y)}[\ell(h(z), y)] - \mathbb{E}_{P_{R'''}(z,y)}[\ell(h(z), y)] \leq \underbrace{\tau \, W_d(P_{R'''}, P_T)}_{\textit{Distribution mismatch}}$, with d as in Eq. 2.*

*Proof.*

$$
\begin{aligned}
& \mathbb{E}_{P_T(z,y)}[\ell(h(z), y)] - \mathbb{E}_{P_{R'''}(z,y)}[\ell(h(z), y)] \\
= \; & \mathbb{E}_{P_T(y)}\mathbb{E}_{P_T(z|y)}[\ell(h(z), y)] - \mathbb{E}_{P_{R'''}(y)}\mathbb{E}_{P_{R'''}(z|y)}[\ell(h(z), y)] \\
= \; & \mathbb{E}_{P_T(y)}[\mathbb{E}_{P_T(z|y)}[\ell(h(z), y)] - \mathbb{E}_{P_{R'''}(z|y)}[\ell(h(z), y)]] \; (\text{since } P_T(y) = P_{R'''}(y)) \\
\leq \; & \mathbb{E}_{P_T(y)}[\sup_{\ell' \circ h' \in \tau - Lipschitz} \mathbb{E}_{P_T(z|y)}[\ell'(h'(z), y)] - \mathbb{E}_{P_{R'''}(z|y)}[\ell'(h'(z), y)]] \\
= \; & \mathbb{E}_{P_T(y)}[\tau \, W_{\|\cdot\|_2}(P_T(z|y), P_{R'''}(z|y))] \; (\text{Kantorovich} - \text{Rubinstein duality}) \\
= \; & \tau \, W_d(P_{R'''}, P_T) \; (\text{Lemma 4}).
\end{aligned}
$$

$\square$

## A.4  Final bound (Sec. 3.3)

**Theorem 3.** *Let $\ell$ be the cross entropy loss then under the assumptions of Theorems 1 and 2 we have,*

$$
\mathbb{E}_{P_T(z,y)}[\ell(h_T(z), y)] \leq \underbrace{\mathbb{E}_{P_R(z,y)}[C(y)\ell(h_R(z), y)]}_{\textit{Re-weighted reference task loss}} + \underbrace{H(\mathcal{Y}_{R''}|\mathcal{Y}_{R'})}_{\textit{Label mismatch}} + \underbrace{\tau \, W_d(P_{R'''}, P_T)}_{\textit{Distribution mismatch}}.
$$

*Proof.* Let $\ell \circ h_T$, $\ell \circ h_R$, and $\ell \circ h_{R'''}$ be $\tau-$Lipschitz (from Assumption 1).

$$
\begin{aligned}
\mathbb{E}_{P_T(z,y)}[\ell(h_T(z),y)] &\leq \mathbb{E}_{P_T(z,y)}[\ell(h_{R'''}(z),y)] \text{ (Optimality difference)} \\
&\leq \mathbb{E}_{P_{R'''}(z,y)}[\ell(h_{R'''}(z),y)] + \tau \, W_d(P_{R'''}, P_T) \text{ (Theorem 2)} \\
&\leq \mathbb{E}_{P_R(z,y)}[C(y)\ell(h_R(z),y)] + H(\mathcal{Y}_{R''}|\mathcal{Y}_{R'}) + \tau \, W_d(P_{R'''}, P_T) \text{ (Theorem 1)}.
\end{aligned}
$$

$\square$

In our experiments, we enforce the $\tau-$Lipschitz constraint for $\ell \circ h_R$ and $\ell \circ h_T$ and verify that the Lipschitz constant of $\ell \circ h_{R'''}$ remains close to $\tau$ within tolerance.

### A.5 Extension to non-linear classifiers and non-linear transformations

To extend our analysis to non-linear classifiers/transformations, we allow $A : \mathcal{Z} \to \mathcal{Z}$ to be a non-linear map and the classifiers $h \in \mathcal{H}_{\text{non\_linear}}$ to be also non-linear (such as multi-layer perceptron). In addition to the Assumption 1 (1), which requires $\ell \circ h_{R'''}$ to be $\tau-$Lipschitz, we also require that $h_{R'''}$ belongs to the same class $\mathcal{H}_{\text{non\_linear}}$ as $h_T$ and $h_R$ for any $A$. This holds for example when $h$ is linear and $A$ is also linear or when $h$ is a multilayer perceptron and $A$ is linear. With this additional assumption, all of our proofs work for any linear/non-linear transformation of the feature space, without any change. Thus, Theorem 1, Theorem 2 and Theorem 3 hold for non-linear classifiers as well. With these extensions, our bounds can be used to explain transferability even with non-linear classifiers.

## B   Additional related work

**Distributional divergence-based analyses of learning with distribution shifts (under same feature and label sets):**  Here we review some of the previous works that analyzed the problem of learning under distribution shifts in terms of distributional divergences such as the Wasserstein distance. These analyses apply when the feature and label spaces remain the same between the original and the shifted distribution.

Early works [8, 52, 36] showed that the performance on a shifted distribution (target domain) can be estimated in terms of the performance of the source domain and the distance between the two domains' marginal distributions and labeling functions. Specifically, [8] showed that that

$$\mathcal{E}_T(h, f_T) \leq \mathcal{E}_S(h, f_S) + d_1(P_S, P_T) + \min\{\mathbb{E}_{P_S}[|f_S(z) - f_T(z)|], \mathbb{E}_{\mathcal{D}_T}[|f_S(z) - f_T(z)|]\},$$

where $d_1$ denotes the total variation distance, $f : \mathcal{Z} \to [0,1]$ denotes the labeling function, $h : \mathcal{Z} \to \{0,1\}$ denotes the hypothesis and $\mathcal{E}_P(h, f) := \mathbb{E}_{z \sim P}[|h(z) - f(z)|]$ denote the risk of the hypothesis $h$. A follow up work [52], showed a similar result using type-1 Wasserstein distance for all $K-$Lipschitz continuous hypotheses i.e.,

$$\mathcal{E}_T(h, f_T) \leq \mathcal{E}_S(h, f_S) + 2K \cdot W_1(P_S, P_T) + \lambda,$$

where $\lambda$ is the combined error of the ideal hypothesis $h^*$ that minimizes the combined error $\mathcal{E}_S(h, f_S) + \mathcal{E}_T(h, f_T)$. Another recent work [32] used a target transformation-based approach and Wasserstein distance to quantify learning in the presence of data and label shifts. Other works [31, 50] also presented an analysis based on Wasserstein distance to understand how the accuracy of smoothed classifies and robustness change in the presence of distribution shifts. Compared to these works the bound proposed in Theorem 2 considers cross-entropy loss (which is a popular choice of the loss function in the classification setting) and uses a joint feature and labels Wasserstein distance rather than only marginal Wasserstein distance. These differences make the bound proposed in Theorem 2 useful in the analysis of transfer learning than those proposed in previous works when we have access to labeled target domain data.

**Comparison with [56]:** The closest work to ours is that of [56], which showed that transferability to a target task can be related to transferability to another task (source task in their work) and the label mismatch between the two tasks. However, the bound is proposed in a restrictive setting when both the tasks have the same features but different labels (i.e. same images labeled differently between the two tasks). In this setting, [56], showed that transferability is upper bounded by loss of the source classifier on the source task and the conditional entropy (CE) of the label sets of the two tasks. We

significantly extend this analysis to general tasks which is the most commonly used setting in practice (e.g., our analysis allows us to study transfer learning from ImageNet with 1000 classes to CIFAR-100 with 100 unrelated classes, where both tasks have different images). In this setting, our main result in Theorem 3, shows that transferability involves additional terms such as the distribution mismatch term (in the form of Wasserstein distance), the prior mismatch term (in the form of re-weighted source loss) and the conditional entropy between the label sets. Moreover, the bound proposed by [56] is a special case of our bound with $C$ being the vector of all ones (no prior change) and $A$ being Identity (data distributions of two tasks are the same).

**Comparison with other transferability estimation and model selection methods:** The problem of transferability estimation has gained a lot of attention recently, especially due to the availability of a larger number of pre-trained models. Earlier works used models end-to-end fine-tuned on target tasks to evaluate transferability via task-relatedness estimated using the actual target loss [62] and the Fisher information matrix [1]. However, the requirement of models trained on target tasks limits their practical utility. Other works [22, 21, 54] used the representation of the data from the tasks from these end-to-end fine-tuned models and developed similarity scores that achieved high correlation with transferability. However, these approaches are not practical since they require models end-to-end fine-tuned on the target task. Our work does not depend on models trained on target tasks as shown in our analysis (Theorem 3) making it both theoretically and practically sound.

Another line of work [5, 40, 29, 61, 55, 51, 19] focuses on the problem of pre-trained model selection where the goal is to find the best pre-trained classifier from a model zoo that will achieve the highest transferability to a particular downstream task. Thus, the main challenge of this problem is to be able to estimate transferability in a way that is more efficient than fine-tuning the pre-trained models on the downstream tasks. To this end, recent works have proposed several scores that are correlated with the accuracy of the models after fine-tuning them on the target task, which we refer to as score-based transferability estimation (SbTE) methods, in our work.

However, unlike our work, the goal of SbTE works is not to propose a universal bound or identify terms that provably govern transferability. Moreover, while the scores proposed in SbTE works correlate well with transferability, they are only meaningful in a relative sense. Concretely, a score of 1 (e.g., for LogMe [46]) on a CIFAR-100 task for a particular model does not indicate whether transferability is good or bad and requires comparison with scores of other pre-trained models on the same target task. On the other hand, our upper bound directly approximates transferability, e.g., an upper bound of 1 on the CIFAR-100 task for a model implies that transfer learning will incur an average cross-entropy loss of less than 1 implying a highly accurate transfer. Our results in Figs. 3, and 9 attest that our upper bound is indeed a good estimate of the transferability.

Another disadvantage of the scores proposed in these works is that they cannot be compared across target tasks, unlike our upper bound. As observed from Fig. 4 of LogMe [46], scores for CIFAR-10 are lower than scores for CIFAR-100 on the same pre-trained model, but, the transferability to CIFAR-10 is better than that to CIFAR-100. On the other hand, from our Fig. 9, the upper bounds on CIFAR-10 are lower than those of CIFAR-100 implying better transferability of classifiers pre-trained on ImageNet to CIFAR-10. Thus, our work is more suitable for estimating the absolute performance on various target tasks given a particular pre-trained classifier.

Thus, while the goal of our work differs fundamentally from SbTE approaches the problem addressed in this work is of significant practical importance. In Sec. 4.3, we show that task-relatedness estimated via Alg. 1 achieves competitive performance compared to popular SbTE methods on this problem.

**Comparison with task transfer learning approaches:** This approach focuses on explaining the transfer performance based on the relationship between tasks. For instance, works such as [28, 57, 24] study the problem of few-shot learning (FSL) where a model is trained on data from related training tasks and is adapted to an unseen task using only a few samples. Different from these works, we focus on the setting where a pre-trained encoder trained on some pre-training dataset is adapted with enough samples/gradient steps to a downstream task. This downstream target task may or may not have any relation to the pre-training task unlike [28, 57, 24].

Concretely, [28] proposed a model-agnostic metric called Task Attribute Distance to gauge the success of transfer. Our work, on the other hand, defines task-relatedness based on the similarity of the representations of reference and target tasks in the representations of the pre-trained model (and is model dependent) rather than relying on the attribute information, which may not be available in TL

setting. [57] analyzes the sample complexity for learning a model shared across tasks and adapting it to a new target task and shows task diversity to be a crucial component for the success of transfer in the FSL setting. Our work on the other hand does not assume access to shared tasks or restrict the number of samples required for fine-tuning on the target task. Moreover, their notion of task diversity requires access to a set of training tasks that may not be available in the TL setting, making our notion of task-relatedness more practical for TL. Lastly, [24] proposes a notion of task-relatedness for the FSL setting, allowing to utilize all the data from available training tasks to help learn a model on a new task with a few gradient steps. This notion is model-agnostic and defined over the sample space ($\mathcal{X} \times \mathcal{Y}$) unlike our measure which is defined in the representation space of the model whose transferability needs to be evaluated.

Thus while task-relatedness is at the core TL, the notions proposed by [28, 57, 24] are suitable for task transfer setting where as our notion is more suitable for the inductive TL setting.

## C  Additional experiments

### C.1  Small-scale experiment to evaluate Alg. 1

*1) Our algorithm produces transformations that achieve a better value for the bound compared to naive transformations. 2) Having the same number of classes in the reference task as in the target task leads to the best value of the bound. 3) It is only marginally better to have semantically related classes in the reference task.*

We evaluate Alg. 1 for computing task-relatedness and predicting the transferability of a ResNet-18 model to CIFAR-10 in two settings. In the first setting, we consider a reference task that comprises data from 10 classes chosen at random and 10 classes that are semantically related to the labels of CIFAR10 from ImageNet [17] (see App. D) whereas in the setting we select reference task with 20 classes. We compare the cross-entropy loss on CIFAR10 obtained after linear fine-tuning (transferability) with our bound, by using various transformations including those learned via Alg. 1.

We test 3 different cases. The first is **FixedAll:** where all transformations are fixed ($A$ is fixed to the Identity matrix, $B$ is fixed to a random permutation matrix in the setting with 10 classes, and to a matrix such that two reference classes match a single target class in the setting with 20 classes, $D$ is fixed to the reference prior). The second is **LearnedA:** where we use Alg. 1 to learn only $A$ while $B, D$ are fixed as in FixedAll, and lastly **LearnedAll:** where all the transformations are learned via Alg. 1. Our results in Fig. 6, show that in both settings using FixedAll, our bound is marginally better when the reference task has classes semantically related to the target task. In all cases, the bound becomes significantly better in the LearnedA setting due to learning the transformation $A$ that transforms the reference distribution to better match the target and decrease the distribution mismatch term of the bound. Lastly, by learning all the transformations, in the LearnedAll setting, we achieve the best value for the bound, regardless of whether there are randomly chosen or semantically related classes in the reference task. Moreover, we see that the value of the bound estimated when the reference task has 10 source classes is better in all cases. This is due to smaller re-weighted reference loss since learning a 20-way classifier is more challenging than learning a 10-way classifier, especially with the Lipschitz constraint. In the setting with 20 classes, Alg. 1, prefers

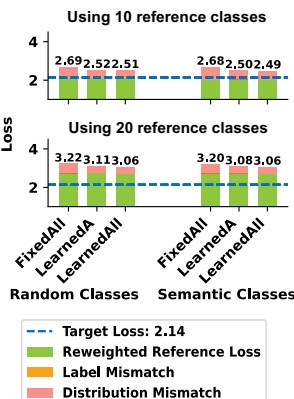

Figure 6: Transformations optimized using Eq. 4 produce a better task-relatedness than obtained by naively chosen transformations, primarily due to decreased distribution mismatch. The presence of the same number of classes in the reference as those in the target achieves the smallest task-relatedness value. Semantically related classes in the reference task help but only marginally.

to retain data from 10 of the 20 reference task classes, Fig. 7 (right), reducing the re-weighted reference loss, makes $B$ sparse, reducing the label mismatch term, and aligns the reference and target distributions, via $A$, reducing the distribution mismatch term.

Overall, our results show that 1) learning $D$ prefers to retain the data from the same number of reference classes as those present in the target, 2) the $B$ matrix eventually becomes sparse, making the label mismatch term zero, and 3) there is only a small difference in task-relatedness between LearnedA and LearnedAll settings. Based on these insights, we use Alg. 1 in the *LearnedA* setting, with the reference task containing the same number of randomly sampled classes from ImageNet as the same number of classes in the target task (since it may not always possible to find semantically similar classes for all target tasks) and fix $B$ to be a random permutation matrix, for all other experiments in the paper. We select classes for the reference task from ImageNet due to its diversity and the fact that it has a large number of classes. However, any dataset can be used to define the reference task (e.g., we used Digits datasets for the reference task in Sec. 4.2).

Next, we evaluate the effect of knowing the exact matching between the labels of the reference and the target tasks in comparison to a random permutation. We use MNIST as the reference task and USPS as the target task (and vice-versa). We compare our results in a setting where only $A$ is learned and $B$ is set to an identity matrix and when $B$ is set to a random permutation matrix. Note that the identity matrix corresponds to the correct mapping between the classes of MNIST and USPS tasks (both contain digits from 0 to 9).

We find that the upper bound obtained when $B$ is fixed to identity is only marginally better than the case when $B$ is a random permutation. Specifically, the difference between the bound when $B$ is fixed to a random permutation and when $B$ is an identity matrix is 0.10 for the MNIST→USPS task and 0.17 for the USPS→MNIST task. The primary reason for the decrease in the upper bound comes from the reduced distribution mismatch term. While the upper bound improves slightly when the ideal matching between the labels is known, such a mapping may not be known when the labels of the tasks are not related such as for FMNIST and MNIST. Thus, fixing $B$ to a random permutation matrix yields a reliable estimate of transferability in most cases.

### C.1.1 Visualization of the transformed data via t-SNE for various settings in Sec. C.1

Here we use the setting considered in App. C.1 where we consider 20 randomly selected classes from ImageNet as the reference task and consider the transfer to CIFAR-10. We plot the results of using different transformations using t-SNE to show how various transformations affect the upper bound in Theorem 3. Our results in Fig. 7(left) show that when no transformations are learned (FixedAll), the 20 random reference task classes do not overlap with the 10 target classes leading to an increased Wasserstein distance which in turn leads to a larger upper bound. By learning the transformation $A$ (LearnedA), Fig. 7(center) shows a significantly better alignment between the classes of the reference and target which leads to a decreased Wasserstein distance and hence a tighter upper bound. Moreover, by learning all the transformations (LearnedAll), Fig. 7(right) shows that not only do the distributions align well but also the prior of the reference is changed to only keep 10 reference classes to match the prior of the target distribution providing a further improvement in the upper bound. This clearly shows the effectiveness of our proposed optimization algorithm in learning various transformations to minimize the upper bound.

### C.1.2 Effectiveness of minimizing the upper bound in Theorem 3 via solving Eq. 4

In Fig. 8, we show how the upper bound changes as the optimization progresses to transform the reference task (ImageNet) into four target tasks with the ResNet-18 model. Similar to experiments in Sec. 4.1 of the paper we optimize over the transformation $A$ while $B$ and $D$ are fixed to a random permutation matrix and the reference prior. After about 600 epochs the optimization problem converges to a local minima.

### C.2 Additional results for the impact of the reference task on task-relatedness (Sec. 4.2)

### C.2.1 Additional results for image classification

Here we provide details of the experiment presented in Sec. 4.2 about the effect of the reference task on task-relatedness and transferability. Similar to the results presented in Fig. 4 of the main paper, the results in Fig. 10 show that when the reference and the target tasks are related then task-relatedness is good as in the case when the reference task is MNIST and target is USPS or vice versa achieving a smaller gap to transferability. When the target tasks are unrelated to the reference task data then both the transferability and task-relatedness are low. E.g., when the reference task is MNIST and the

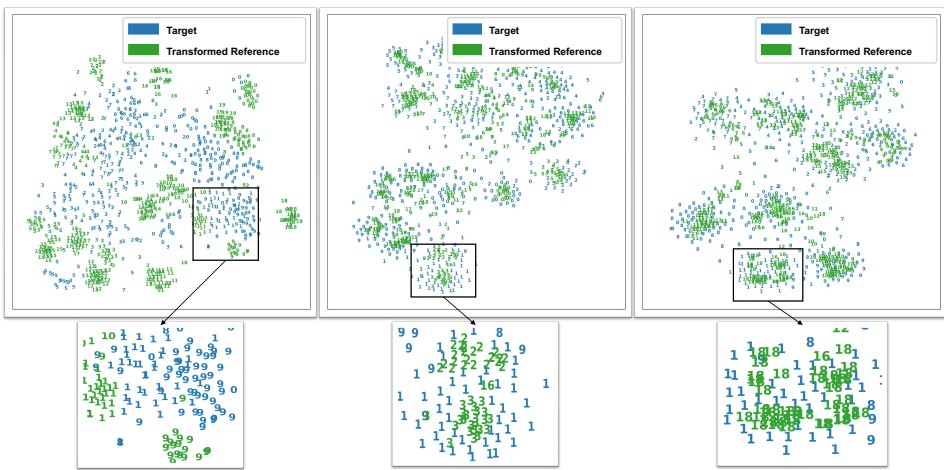

Figure 7: (Best viewed in color.) t-SNE visualizations of the effect of various transformations on the bound in Theorem 3 when data from 20 randomly selected classes from ImageNet are used to transfer to CIFAR-10. When all transformations are fixed (FixedAll, left) the distance between the distribution $R'''$ (transformed reference) and $T$ is high explaining the large upper bound. Learning just the transformation $A$ using the algorithm proposed in Sec. 3.4 significantly reduces the distance between $R'''$ and $T$ leading to a tighter upper bound (center). Learning all the transformations further improves the matching (right). Especially, learning $B$ and $D$ change the class priors of the reference so that the same number of classes from the reference are used for matching as those present in the target. This is evident from the right plot where only 10 unique reference task clusters are visible compared to 20 in the center plot, with fixed D. Moreover, the zoomed-in portion shows that for the center figure two classes from the reference task (green 2 and 3) match with class 1 (blue) of the target whereas a single class from the reference task (green 18) matches class 1 (blue) of the target in the right figure.

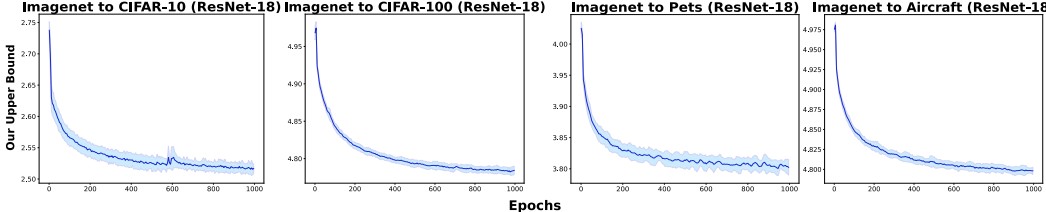

Figure 8: Reduction of the proposed upper bound is shown as transformations are learned by solving the optimization problem in Eq. 3. After 600 epochs, the upper bound stabilizes showing the convergence of the optimization problem. Each subplot shows the effect of learning the transformation parameters for the transfer learning task with ImageNet as the reference task and ResNet-18 (trained in a supervised way) for different target tasks. The solid line is the average after 5 random restarts and the shaded portion shows their standard deviation.

target is FMNIST or vice-versa. Task-relatedness is also correlated to transferability, i.e., a model trained on MNIST achieves better transferability to USPS than FMNIST.

### C.2.2  Results for NLP sentence classification task

In this section, we use sentence classification NLP task to demonstrate further the effect of the reference task on task-relatedness. For this experiment, we first fine-tune the entire DistilRoBERTa [34] model distilled on OpenAI's WebText dataset, using a subsample of 10,000 points from the DBPedia dataset. We then use these fine-tuned models to evaluate the transferability to AG news, SST-5, and Yelp datasets. The results in Fig. 12 show that transferability on AG News is the smallest among the three datasets. This coincides with the task-relatedness value obtained after learning the

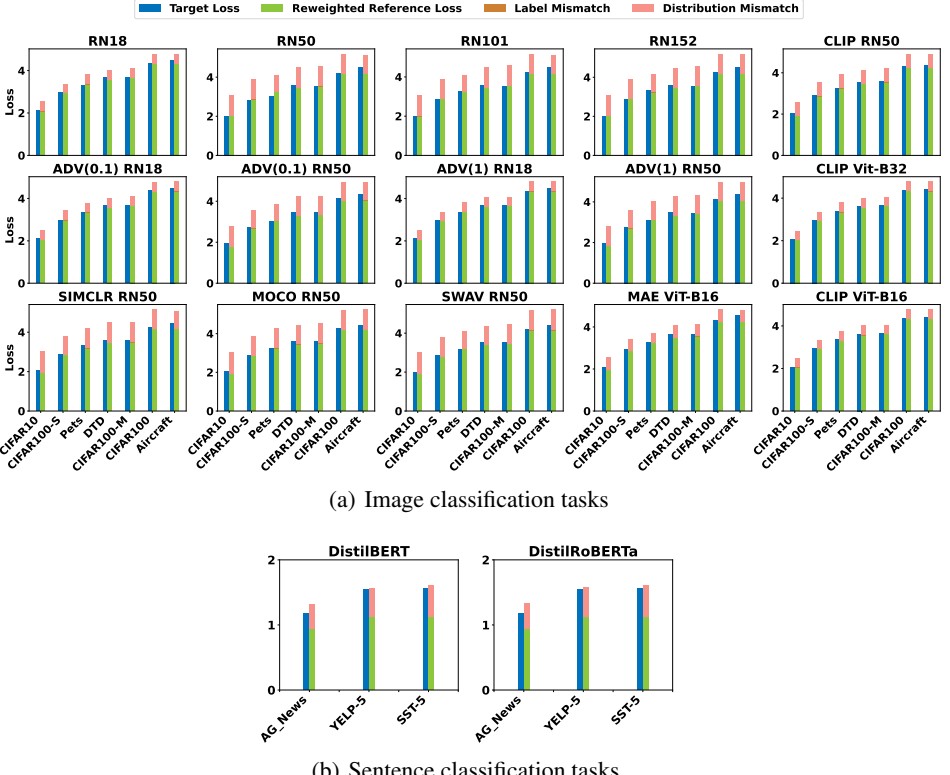

(a) Image classification tasks

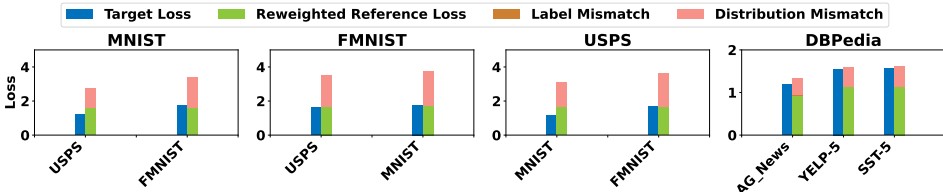

(b) Sentence classification tasks

Figure 9: Full results for comparison of transferability vs. task-relatedness for large pre-trained models on image and sentence classification tasks. Task-relatedness consistently achieves a small gap to transferability. We denote cross-entropy loss on the y-axis. Plot-title denotes the pre-trained model and the x-axis denotes the target tasks.

Figure 10: Decomposition of task-relatedness into its three components illustrates that our task-relatedness (specifically the distribution mismatch term) explains the difference in transferability. Similar tasks such as USPS and MNIST have the highest transferability and also have the highest task-relatedness.

transformations which explains why transfer from DBPedia to AG News is more successful compared to other target tasks. This observation is reasonable, especially considering that both DBPedia and AG News have structured information. Moreover, since DBPedia is related to Wikipedia, the terms and entities appearing in AG News are more related to those appearing in DBPedia in comparison to terms/entities appearing in SST-5 and Yelp which consist of movie reviews and reviews collected from Yelp.

For our experiments, in this section, we follow a similar setting of fixing $B$ to be a random permutation matrix, $C$ to the prior of the reference task, and only learn the transformation $A$. We sample 10,000 points from DBPedia belonging to the same number of classes as those present in the target task (e.g., for AG News we sample data from 4 randomly selected classes of DBPedia) and use this data as the reference task data to train $h_R$ with gradient norm penalty ($\tau = 0.02$). All experiments are run for 3 random seeds and average results are reported in Fig. 12.

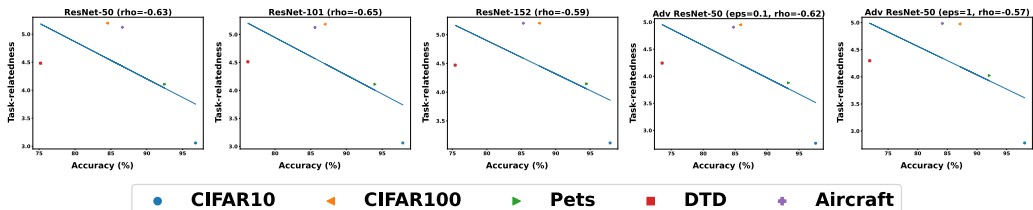

Figure 11: Task-relatedness is highly (negatively) correlated (Pearson correlation coefficient in the subplot title) with the accuracy of the models after end-to-end fine-tuning. Each subplot shows transfer learning with various target tasks for a specific model architecture and training method.

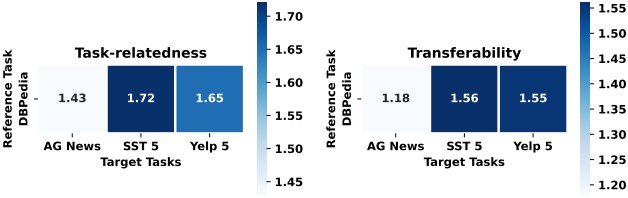

Figure 12: Task-relatedness (Left) and transferability (Right) are highly correlated across various reference-target pairs. A target task related to the reference task (DBPedia) such as AG News achieves better transferability (with DistillRoBERTa model) and task-relatedness compared to less-related tasks such as SST-5 and Yelp5.

### C.3 Lipschitz constrained linear fine-tuning

#### C.3.1 Implementing softmax classification with $\tau-$Lipschitz loss

To use the bound Theorem. 3, it is required that the loss be $\tau-$Lipschitz continuous w.r.t. $z$ in the input domain $\mathcal{Z}$. To enforce this, while learning the weights of the softmax classifier (linear fine-tuning) for the reference task or the target, we add the gradient norm penalty as used in previous works [52, 4] and solve the following optimization problem

$$\min_{h} \frac{1}{N} \sum_i \left[ \ell(h(z_i), y_i) + \rho \max_y \max\{0, \|\nabla_z \ell(h(z_i), y)\|_2 - \tau\}^2 \right] \quad (\rho \approx 10^4)$$

where $\ell(h(z), y) = -w_y^T z + \log \sum_j e^{w_j^T z}$ is the cross-entropy loss.

#### C.3.2 Trade-off between empirical and predicted transferability

Constraining the Lipschitz coefficient of the classifier increases both the target and the reference task cross-entropy loss since the hypothesis set is being restricted. The smaller the $\tau$ is, the larger the loss becomes. On the other hand, the smaller $\tau$ makes the distribution mismatch term in Theorem 3 also smaller. Since the bound is the sum of the reference task loss and the distribution mismatch (and label mismatch), there is a trade-off determined by the value of $\tau$. We illustrate the effect of the values of $\tau$ on the empirical and predicted transferability. As mentioned previously, we train both the classifier for the reference task $h_R$ and the target $h_T$ with an additional penalty on the gradient norm to make them $\tau-$Lipschitz. In Fig. 13, we present results of varying the value of $\tau$ for the transfer to the Pets dataset with ImageNet as the reference task. For this experiment, we selected 37 random classes from ImageNet and only learned the transform $A$ by keeping $B$ fixed to a random permutation and $C$ fixed to the uniform prior over reference task classes. We observe that the performance of linear fine-tuning degrades as we decrease the value of $\tau$ but explainability through the bound improves since the distribution mismatch term (dependent on $\tau$) decreases in the bound. However, making $\tau$ too small is not preferable since it leads to an increase in the first term of the bound (re-weighted reference task loss) increasing the bound overall. Moreover, it also leads to a degradation in the accuracy after linear fine-tuning. For our experiments, we use $\tau = 0.02$ since it doesn't decrease the accuracy of fine-tuning significantly and leads to a small gap between empirical and predicted transferability.

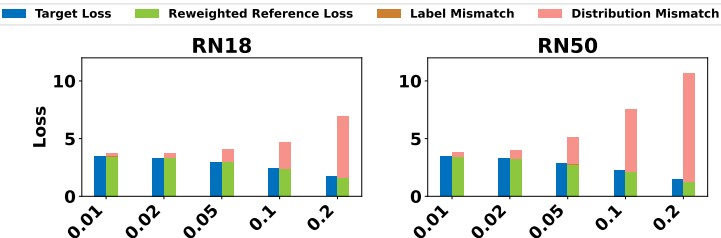

Figure 13: Trade-off between the cross-entropy loss (y-axis) after linear fine-tuning and the upper bound in Theorem 3 as a function of $\tau$ (x-axis), for ResNet18 and ResNet50 models pre-trained on ImageNet and linearly fine-tuned on the Pets dataset. Increasing the value of $\tau$ leads to a decrease in the cross-entropy loss after fine-tuning but increases in the proposed bound mainly due to the $\tau \cdot W_d$ term.

# D   Details of the experiments

All codes are written in Python using Tensorflow/Pytorch and were run on an Intel(R) Xeon(R) Platinum 8358 CPU with 200 GB of RAM and an Nvidia A10 GPU. Implementation and hyperparameters are described below. Our codes can be found in the supplementary material. We report an average of three independent runs for experiments in Sec. 4.2 and 4.3.

## D.1   Dataset details

In our work, we used the standard image classification benchmark datasets along with standard natural language processing datasets[1].

**Aircraft [35]:** consists of 10,000 aircraft images belonging to 100 classes.

**CIFAR-10/100 [30]:** These datasets contain 60,000 images belonging to 10/100 categories. Additionally, we created two subsets of CIFAR100 with the first 25 (small CIFAR-100-S) and 50 (medium CIFAR-100-M) classes.

**DTD[13]:** consists of 5,640 textural images belonging to 47 categories.

**Fashion MNIST [60]:** consists of 70,000 grayscale images belonging to 10 categories.

**Pets [43]:** consists of 7049 images of Cats and Dogs spread across 37 categories.

**ImageNet [17]:** consists of 1.1 million images belonging to 1000 categories.

**Yelp [63]:** consists of 650,000 training and 50,000 test examples belonging to 5 classes.

**Stanford Sentiment Treebank (SST-5) [63]:** consists of 8,544 training and 2,210 test samples belonging to 5 classes.

**AG News [63]:** consists of 120,000 training and 7,600 test examples belonging to 4 classes

**DBPedia [63]:** consists of 560,000 training and 70,000 test examples belonging to 14 classes

## D.2   Semantically similar classes for CIFAR-10 from ImageNet

For our experiments with CIFAR-10 in Sec. C.1, we selected the following semantically similar classes from ImageNet, {`airliner, minivan, cock, tabby cat, ox, chihuahua, bullfrog, sorrel, submarine, fire engine`}.

## D.3   Additional experimental details

Details of the optimization problem in Eq. 4 and Alg. 1. In Step 5 of the algorithm, we use the network simplex flow algorithm from POT [23] to compute the optimal coupling. Since computing the Wasserstein distance over the entire dataset can be slow, we follow [16] and compute the coupling

---

[1]All NLP datasets and models are obtained from `https://huggingface.co/`.

over batches. Note that the base distance defined in Eq. 2 is non-differentiable. Thus, we use a differentiable approximation $\tilde{d}((z, y), (z', y')) := d_{features}(z, z') + \nu \cdot \|e(y) - e(y')\|_2$ (with $\nu = 10^8$) where $(z, y)$ and $(z', y')$ are samples from the domains $R'''$ and $T$ and $e(\cdot)$ denotes the one-hot embedding of the labels in Step 5/6 . The first three terms in Step 6 of our algorithm correspond to the terms in the objective of Eq. 4 while the two additional terms are added to penalize the constraints of class prior matching $P_T(y) = BD$ and invertibility of the matrix $A$, respectively as required by Theorem 3. We use the softmax operation to ensure $B$ and $D$ are a valid probability matrix and vector.

In our experiments, in Sec. 4.1, we used pre-trained models available from Pytorch for ResNet18/50, along with publicly available pre-trained models provided in the official repositories of each training method. For each experiment, we subsample data from the ImageNet dataset belonging to the same number of classes as those present in the target dataset and use this data to train the linear layer on top of the representations extracted from the pre-trained model along with a gradient norm penalty (Reference task classifier). To speed up the experiments, we use only 10,000 points from the subsample of ImageNet for training the linear classifier and computing the transfer. For evaluation, we use a similar subsample of the validation dataset of ImageNet containing all the samples belonging to the subsampled classes. Fine-tuning on this dataset takes about 0.05 seconds per epoch for the task of transfer from ImageNet to Pets with the ResNet-18 model (we run the fine-tuning for a total of 5000 epochs).

Along with training the linear classifiers with a gradient norm penalty (with $\tau = 0.02$), we standardize the features extracted from the pre-trained models to remove their mean (along each axis) and make them have a unit standard deviation (along each axis). While standardizing the features do not have a significant impact on the loss of the classifiers, including it makes it easier to match the distributions of the reference task and target data after transformations. Since our optimization problem transforms the reference task distribution to match the distribution of the target by solving the optimization problem in Eq. 4 by working on mini-batches, the size of the batch must be greater than the dimension of the representation space of the pre-trained encoder. E.g., for ResNet18 models which have a representation dimension of 512, we use a batch size of 1000 and for ResNet50 models which have a representation dimension of 2048, we use a batch size of 2500. Having a smaller batch size than the dimension could lead to a noisy gradient since for that batch the transformation can achieve a perfect matching, which may not generalize to data from other batches or unseen test data.

While computing the transformations, we apply the same augmentation (re-sizing and center cropping)/normalization to the training data as those applied to the test data. Along with this, we extract the features of the training and test data from the pre-trained model once and use these to train the linear layer. We note that this is done to save the computation time and better results could be obtained by allowing for extracting features after data augmentation for every batch.

Finally, for our experiments in Sec. 4.2, the encoders are trained end-to-end on the reference task. This is in contrast to our other experiments where the encoders are pre-trained and data from the reference task is only used for linear fine-tuning. Using these models, task-relatedness is evaluated by fine-tuning a linear layer using the data from the target task as well as the transformations computed by solving Eq. 4. We used $\tau = 0.2$ here. We run the experiments with 3 random seeds and report the average results.

For our experiments in Sec. 4.3, we used the official code from [61] to compute the scores for NCE, Leep, and LogMe along with the official code of [51] for SFDA. For PACTran [19], we also use their official code with the PACTran-Gaussian method with $N/K = 100, \beta = 10N, \sigma^2 = D/100$ where $N$ denotes the number of samples and $K$ denotes the number of classes. This setting is similar to the PACTran-Gauss$_{fix}$ setting used in their work with the difference that we use $N/K = 100$ to use a sufficiently large number of samples, especially considering that all our other results for SbTE methods are computed on the full training set. For OTCE, we follow the official code and compute the recommended OT-based NCE score and OTCE score ($\lambda_1 = -0.0001$ and $\lambda_2 = -1$) using 4000 randomly selected training samples from the two tasks. For the source task, we subsample the same number of classes as the target task and use. For the H-score, we use the official code to compute the H-score [6].

For estimating transferability by solving Eq. 5, we set $\lambda$=0.01 for all the experiments in Sec. 4.3.

