# OpenReview forum: "Understanding the Transferability of Representations via Task-Relatedness"
_NeurIPS.cc/2024/Conference — NeurIPS 2024 poster_

### Official Review · Reviewer_sD5Z · 2024-07-07

**Soundness:** 2
**Presentation:** 3
**Contribution:** 2
**Rating:** 5
**Confidence:** 2

**Summary:**

This paper considers transfer learning in a general cross-domain, cross-task setting. It introduces the concept of "task-relatedness" into measuring whether transfer learning can be successful. Specifically, task-relatedness consists of (1) a reference loss term for measuring the difference in class prior distribution between the target and source tasks; (2) a label mismatch term for measuring the conditional entropy between the label distributions; and (3) a distribution mismatch term  for measuring the difference in feature distributions. To make computing task-relatedness practical, the paper proposes several approximation schemes. It shows the metric is highly correlated with model transferrability in practice and can be used for selecting pretrained models.

**Strengths:**

1. Practical significance: The topic studied is important to the field of machine learning, as fine-tuning large-scale pretrained models gradually becomes the to-go option for model development. The idea of decomposing transferability into prior, feature, and label distribution differences is clean and interesting.
2. Complete pipeline: the paper not only proposes task-relatedness as a transferability metric but also shows how it can be transformed into a training objective for effective transfer learning. This makes the paper more comprehensive and the proposed method more valuable.
3. Presentation is clear and code for reproducing the experiments is provided

**Weaknesses:**

1. Limited novelty: the idea of using distribution distances to measure transferability is not new. As the related work section has pointed out, using the Wasserstein distance to estimate domain difference has been studied by OTDD [1]. Works like OTCE [2] also uses the proposed metric to do pretrained model selection. One argument the paper makes is that the proposed task-relatedness works without requiring target labels. While this does add some practical significance, the proposed method is a simple pseudo-labeling scheme without theoretical justification.
2. Limited evaluation: the baselines considered in the experiments are not comprehensive, e.g., OTCE is not evaluated, and other recent OT-based works like [3] are not discussed. There's also no comparison for different pseudo-labeling schemes for the missing-label setting. Besides, there's no ablation study showing the contribution of each of the three terms proposed for computing task-relatedness.

[1] Geometric Dataset Distances via Optimal Transport. David Alvarez-Melis. Nicolò Fusi.

[2] OTCE: A Transferability Metric for Cross-Domain Cross-Task Representations. Yang Tan, Yang Li, Shao-Lun Huang.

[3] Wasserstein Task Embedding for Measuring Task Similarities. Xinran Liu, Yikun Bai, Yuzhe Lu, Andrea Soltoggio, Soheil Kolouri.

**Questions:**

1. Class-prior transformation assumes the reference task has more classes than the target task. Is this too strong an assumption? Does the proposed method applies to the settings where the target task has more labels?

2. In Figure 3, the rankings of transferability for different target tasks are the same across all models. This is perhaps because the same reference dataset is used, and the experiments do not show the effect of varying model backbones. It would be better if the paper can evaluate multiple reference datasets on multiple models. (Figure 4 does look at different reference datasets but the model is fixed here.)

3. Does the proposed metric applyy to non-classification tasks like regression or auto-regressive prediction?

4. What's the max cross-domain distance the theory holds? I’d imagine that when the domain gap is large, the theory breaks. Have you studied that?

**Limitations:**

See weaknesses.

---

> ### Author Rebuttal · Authors · 2024-08-05
>
> Thank you for your thoughtful review. We are delighted that you found our analysis clear and method practically valuable. We respond to your concerns below:
>
> > 1. Limited novelty: (A) the idea ... is not new. (B) the proposed method ... justification.
>
> A: While we agree with the reviewer that distribution distance is effective at gauging transferability, prior works such as OTDD and OTCE offer limited theoretical justification to explain a model’s transferability. As mentioned by the reviewer in their paper’s summary, our analysis shows that there are two additional terms (namely, reweighted reference loss and conditional entropy of label sets) along with distribution distance to provably explain transferability. Moreover, unlike OTDD and OTCE which rely on the distance between the distributions of the reference and the target tasks, in the input and representation space, respectively, our analysis shows that the distance between the **transformed** distribution of the reference task in the representation space (i.e. P_{R’’’} obtained after using the transformation A) and the distribution of the target task is what explains transferability provably (Theorem 3).
>
> B: We agree with the reviewer that task-relatedness can estimate transferability without requiring target labels easily and we emphasize that this ability is unique to our approach. This is because, to the best of our knowledge, no other transferability estimation method can produce pseudo-labels of the target data in the label space of the **target** task (Methods such as LEEP [34] rely on pseudo-labels of the target data in the label space of the **source** task). Using these labels for the target task, we can compute task-relatedness as per Def. 2 and still estimate the transferability of the models (Tabel 2). This makes task-relatedness the only method that can gauge the transferability of models without access to target labels.
>
> Thus, compared to prior works, our paper advances both our theoretical and practical understanding of transfer learning in a meaningful way.
>
>
> > 2. Limited evaluation: (A) OTCE is ... not discussed. (B) no comparison ... pseudo-labeling schemes (C) no ablation study ... task-relatedness.
>
> A: Please see our joint response for comparison with OTCE.
>
> We thank the reviewer for pointing us to [a] and we briefly compare our approach to them here. We will include this discussion in the main paper.
>
> [a] proposes a model-agnostic approach that relies on optimal transport to compute the distance between the tasks similar to OTDD [3]. However, our work specifically focuses on understanding the transferability of different models both theoretically and practically and hence proposes a model-dependent metric for it.
>
> B: As mentioned above, no previous score-based transferability estimation methods can work without access to labels of the target data. Methods such as LEEP [34], use pseudo-labels of the target data from the source classifier but still require labels of the target data to compute their transferability score. Thus, to the best of our knowledge, task-relatedness is the only approach that can work when no target labels are available. Due to this, we show its effectiveness by comparing the correlation of task-relatedness with and without target labels to actual transferability in Table 2. Our results show that even without target labels, task-relatedness achieves a good correlation with transferability when reference and target tasks are related.
>
> C: In Appendix C.1, we study the effect of different transformations on task-relatedness. We present three settings, the first where no transformations are learned, the second where feature transformation is learned, and the third where all the transformations are learned. Our results show that task-relatedness achieves the smallest gap to transferability when all transformations are learned, closely followed by learning only the feature transformation, and significantly better than not learning any transformation. These three settings capture the effect of the three different transformations proposed in our transformation model. We also break down task-relatedness into its three corresponding terms in Figs. 3 or 4b to highlight the contribution of each term.
>
> > 1. Class-prior ... assumption?
>
> Please see our joint response.
>
> > 2. In Figure 3, ... fixed here.)
>
> The full results of Fig 3 presented in Fig 9 (in the Appendix) already include models trained with different model backbones such as ResNet-18, ResNet-50, ResNet-101, and ResNet-152 trained with a variety of training algorithms. Along with these results, Fig 4(a) shows the effect of training the same model backbone on different reference datasets and evaluates transferability to a variety of target tasks. Fig 4(b), evaluates transferability to various target tasks using the original pre-trained CLIP encoder and CLIP encoder fine-tuned end-to-end on different reference tasks. Thus, our experiments already evaluate the effect of different model backbones and reference datasets on transferability and task-relatedness.
>
> > 3. Application ... auto-regressive prediction?
>
> Please see our joint response.
>
> > 4. What's the ... holds?
>
> Similar to any other upper bounds that analyze the problem of distribution shift (e.g., [6]), a large upper bound implies that theory might not be able to explain the results in practice. This is true for task-relatedness as well since a large upper bound does not necessarily guarantee poor transferability. Using various datasets/tasks (in Fig 3 and 9) we show that task-relatedness estimates transferability closely. Moreover, under the conditions mentioned in Lines 221-225, our bound is indeed tight.
>
> **New References**
>
> [a]: Wasserstein Task Embedding for Measuring Task Similarities

---

> > ### Comment · Reviewer_sD5Z · 2024-08-12
> >
> > I appreciate the authors' detailed response. I'll raise my score to a 5.

---

> > > ### Author Response · Authors · 2024-08-12
> > > **Response by Authors**
> > >
> > > We appreciate the reviewer's response to our rebuttal. We would be happy to address any other concerns the reviewer has.

---

### Official Review · Reviewer_dEJE · 2024-07-08

**Soundness:** 2
**Presentation:** 3
**Contribution:** 3
**Rating:** 5
**Confidence:** 3

**Summary:**

This paper proposes an analysis that analyzes the transferability of the representations of pre-trained models to downstream tasks in terms of their relatedness to a given reference task. It aims to understand when the knowledge of these pre-trained models can be transferred to obtain high-performing models on downstream target tasks. Their analysis leads to an upper bound on transferability in terms of task-relatedness, quantified using the difference between the class priors, label sets, and features of the two tasks. The experiments results demonstrate the effectiveness of the proposed method.

**Strengths:**

(i) This work analyzes transferability for classification tasks and provides the first upper bound on transferability in terms of task-relatedness in a cross-domain cross-task setting.

(ii) This work proposes an optimization problem to compute task-relatedness using a small amount of target labels and shows that it can even predict performance after end-to-end fine-tuning without requiring target labels.

(iii) This work conduct multiple experiments to demonstrate the effectiveness of the proposed method.

**Weaknesses:**

(i) This work mentioned that it is the first to analyze transferability for classification tasks under cross-domain cross-task settings. However, there are multiple works [1-5] that have focused on the transferability and generalization of learned representations. Meanwhile, transferability is mainly evaluated by utilizing the performance differences between different domains and tasks with distribution shifts. This is also the case in previous works, but this is not the case for the introduction of motivation in this paper.\
[1] Understanding few-shot learning: Measuring task relatedness and adaptation difficulty via attributes.\
[2] Otce: A transferability metric for cross-domain cross-task representations.\
[3] On the theory of transfer learning: The importance of task diversity.\
[4] An information-theoretic approach to transferability in task transfer learning.\
[5] Task relatedness-based generalization bounds for meta learning.

(ii) (Minor) Although this work focuses more on theoretical exploration, empirical verification on real-world datasets is also necessary, such as ablation studies on the three proposed terms, trade-off experiments (performance vs. efficiency vs. memory footprint) due to the introduction of additional computations, and comparisons with baselines on transferability.

(iii) (Minor) The formatting and layout of some formulas and tables need to be adjusted to make the reading clearer and more intuitive while some equations with writing errors, such as Eq. 3, L527-528, the proof of Corollary 1. At the same time, due to the large number of formulas and physical quantities, it may be better to construct a table to explain the symbols and definitions.

(iv) (Minor) The track chosen by the paper is not that matched to its content.

**Questions:**

Please see Weaknesses.

---

> ### Author Rebuttal · Authors · 2024-08-05
>
> Thank you for your thoughtful review. We respond to your concerns below:
>
> > Compare with [1,2,3,4,5]
>
> We thank the reviewers for pointing us to additional related works. We briefly compare our approach to them here and will include the discussion in the main paper.
>
> [1,3,5]: study the problem of few-shot learning (FSL) where a model is trained on data from related training tasks and is adapted to an unseen task using only a few samples. Different from these works, we focus on the transfer learning (TL) setting where a pre-trained encoder trained on some pre-training dataset is adapted with enough samples/gradient steps to a downstream task. This downstream target task may or may not have any relation to the pre-training task unlike [1,3,5].
>
> Concretely, [1] proposed a model-agnostic metric called Task Attribute Distance to gauge the success of learning in the FSL setting. Our work, on the other hand, defines task-relatedness based on the similarity of the representations of reference and target tasks in the representations of the pre-trained model (and is model dependent) rather than relying on the attribute information, which may not be available in the TL setting.
>
> [2] analyzes the sample complexity for learning a model shared across tasks and adapting it to a new target task and shows task diversity to be a crucial component for the success of FSL. Our work on the other hand does not assume access to shared tasks or restrict the number of samples required for fine-tuning on the target task. Moreover, their notion of task diversity requires access to a set of training tasks that may not be available in the TL setting, making our notion of task-relatedness more practical for TL.
>
> [3] proposes a notion of task-relatedness for the FSL setting, allowing to utilize all the data from available training tasks to help learn a model on a new task with a few gradient steps. This notion is model-agnostic and defined over the sample space ($X \times Y$) unlike our measure which is defined in the representation space of the model whose transferability needs to be evaluated.
>
> Thus while task-relatedness is at the core of both TL and FSL, the works [1,3,5] proposed notions relevant to the FSL setting whereas our work proposed a notion relevant to the TL setting.
>
> Please see the joint response empirical comparison against [2,4].
>
> > Although this work ... baselines on transferability.
>
> **Ablation studies:** In Appendix C.1, we show the effect of using different transformations on the value of task-relatedness computed by solving Eq. 3. We present the results in three settings, the first where none of the transformations are learned, the second where only the feature transformation (parameterized by A) is learned, and the third where all the transformations are learned. The experimental results show that task-relatedness achieves the smallest gap to transferability when all transformations are learned, closely followed by learning only the feature transformation, and significantly better than not learning any transformation. These three settings capture the effect of the three different transformations proposed in our transformation model.
>
> **Trade-off experiments:** In Appendix C.1.2, we evaluate the number of epochs required by Alg 1 to minimize the proposed upper bound (compute task-relatedness) for four target tasks using the ResNet-18 model and Imagenet as the reference task. Our results in Fig 8 show that after approximately 600 epochs (approximately 2 minutes of wall clock time on our hardware) Alg. 1 learns the transformations that lead to a good estimate of transferability.
>
> For the problem of end-to-end transferability estimation (Sec 4.3) for different models, we find that the computation of task-relatedness is significantly faster than the computation of actual end-to-end fine-tuning. Specifically, the computation of task-relatedness requires roughly only 3-4 minutes for computation, compared to end-to-end fine-tuning which can require computation of several hours. E.g., You et al., 2021 [54] show that end-to-end fine-tuning of a ResNet-50 model on the Aircraft dataset, with hyperparameter search, requires about a day’s worth of computation to achieve the best accuracy of 86.6% (see Sec 5.5 and Table 5 of You et al., 2021). In comparison, task-relatedness can be estimated very efficiently. Compared to other SbTE approaches such as SFDA, our approach increases the computational time by ~30 seconds but gets consistently better results as shown in Table 1 and Fig 5 of the paper and the pdf in the joint response.
>
> **Comparisons with baselines on transferability:** In Fig 3 and 4(b), we compare the estimate of transferability computed via Alg. 1 to the ground truth value of transferability (denoted by the blue bar labeled target loss). Our results show that task-relatedness incurs a small error compared to actual transferability.
>
> In Table 1, we present the result of comparing the correlation of task-relatedness and accuracy after end-to-end fine-tuning and compare it to the correlation of five popular score-based transferability estimation methods. Along with these we also compare the effect of the number of target labels available on the correlation of each of these methods with the accuracy after end-to-end fine-tuning.
>
> Thus, we believe that we have already provided a detailed empirical evaluation of our methodology and compared our approach with popular transferability methods. However, if the reviewer would like to see any specific experiments to be added, we would be happy to provide them as well.
>
> > The formatting ... definitions.
>
> We thank the reviewer for this excellent suggestion. We will create a table to clarify all the symbols and their meaning. We will expand and adjust the tables and equations to be more readable in the camera-ready version.

---

> > ### Comment · Reviewer_dEJE · 2024-08-12
> >
> > Thank you for the responses. But my main concerns still exist, for example, the author mentioned that transfer learning is a setting uniquely considered in this article, but meta-learning can be considered a branch of transfer learning; the concept of task similarity is difficult to distinguish from the one in this article, and the descriptions like "can be used or not" and "may not" is vague. Therefore, the confusion in Weakness 1 is difficult to eliminate from the response. At the same time, by reading the responses of other reviewers, I found some issues that were previously overlooked. Therefore, after carefully checking all the responses, I tend to maintain my score.

---

> ### Author Response · Authors · 2024-08-12
> **Response by Authors**
>
> We thank the reviewer for their response to the rebuttal.
>
> While we agree that transfer learning is a broad research topic, our paper specifically works in the inductive transfer learning setting where the focus is to leverage an inductive bias (a pre-trained model) to improve performance on a target task. Due to this, our analysis specifically proposes a model-dependent way of measuring the relatedness between a reference and a target task. The works such as [1,3,5] provided by the reviewer deal with the problem of task transfer learning. Their goal is to identify the relationship between tasks, regardless of the model, to explain the transfer performance.
>
> E.g., the task attribute distance proposed by [1] is independent of the model being used and only depends on the tasks. (This terminology of inductive and task transfer learning is based on a previous work LogMe[54] which has the same setting as ours.). Thus, the setting of our work differs significantly from the setting in [1,3,5].
>
> We thank the reviewer for bringing this up. We will add a discussion to the paper and clarify the type of transfer learning being studied in the paper along with the mention of the relevant papers from the area of task transfer learning.
>
> We would be happy to address any further concerns the reviewer may have.

---

### Official Review · Reviewer_E8p5 · 2024-07-12

**Soundness:** 4
**Presentation:** 4
**Contribution:** 3
**Rating:** 6
**Confidence:** 4

**Summary:**

This paper analyzes transfer learning from the perspective of task relatedness, a model for transforming from a reference task (pretraining task) to the target task is proposed, which consist of prior transform, label transform, and feature transform.
The task relatedness is then measured as the distribution mismatch of the transformed distribution and the target distribution.
Experiments have shown that the proposed task relatedness tightly upperbounds the transferrability to a range of architectures and datasets.

**Strengths:**

1. The motivation and the derivation of the proposed task relatedness make senses to me.
2. The empirical results have shown the advantage of the proposed task relatedness.

**Weaknesses:**

1. The datasets for evaluation and empirical studies seem to be small in scale, for example, MNIST and CIFAR which is small in the image resolution, or Aircraft and DTD which is small in the number of images.

**Questions:**

The appendix has answered most of my questions.
My only question is that for algorithm 1, does this has guaranteed convergence? and practically how long does it take to reach convergence?


Minor:

A possible related work: Discriminability-Transferability Trade-Off: An Information-Theoretic Perspective, ECCV 2022.

**Limitations:**

The paper has discussed its limitations

---

> ### Author Rebuttal · Authors · 2024-08-05
>
> Thank you for your thoughtful review. We are glad that you found our work intuitive. We respond to your concerns below:
>
>
> > The datasets for evaluation and empirical studies seem to be small in scale, for example, MNIST and CIFAR which is small in the image resolution, or Aircraft and DTD which is small in the number of images.
>
> The datasets used in our work are the ones used by prior works. This was done to make the comparison with previously proposed methods for transferability estimation easier. Moreover, most model backbones used in our work require data to have dimensions 224x224, so we resized images from all datasets including CIFAR to match those dimensions. This shows that our method handles data with a larger resolution.
>
> > My only question is that for algorithm 1, does this has guaranteed convergence? and practically how long does it take to reach convergence?
>
> Similar to other problems in deep learning this is a non-convex minimization problem where SGD converges to a local minima. Fig 8, in Appendix C.1.2  shows how Alg. 1 minimizes the proposed upper bound by learning different transformations of the reference task (ImageNet) into four target tasks (CIFAR-10, CIFAR-100, Pets, and Aircraft) with the ResNet-18 model. After approximately 600 epochs (approximately 2 minutes of wall clock time on our hardware)  the Alg. 1 converges to local minima.
>
> To demonstrate that Alg. 1 converges to a reasonable solution, we consider transfer learning using a 20-class subset of Imagenet as the reference task and CIFAR-10 as the target task. Results in Fig. 6 (bottom left) and Fig. 7 (in the Appendix) provide clear evidence of Alg. 1 finding the transformations that minimize the upper bound.
>
> Specifically, without learning any transformations the upper bound is large (~ 3.22 in Fig. 6) and the reference and target classes do not match (leftmost plot in Fig. 7). However, learning all the transformations reduces the upper bound (3.06 in Fig. 6), suppresses the prior of 10 source classes to match the 10 classes in CIFAR-10 and reduces the Wasserstein distance between the transformed source and target classes (rightmost plot in Fig. 7) illustrating the working and convergence of the optimization problem to a good solution.
>
>
> > Compare with Discriminability-Transferability Trade-Off: An Information-Theoretic Perspective
>
> We thank the reviewers for pointing us to this work. We briefly compare our approach to it here and will include the discussion in the main paper.
>
> The work demonstrated a tradeoff between a model's discriminability and transferability properties as training progresses and proposed a learning framework to alleviate this tradeoff. Our work on the other hand focuses on analyzing the transferability in terms of relatedness of the representations of the reference and target tasks after training is complete.

---

### Official Review · Reviewer_3SCA · 2024-07-14

**Soundness:** 3
**Presentation:** 3
**Contribution:** 3
**Rating:** 6
**Confidence:** 4

**Summary:**

The paper presents a way of assessing the transferability of representations from a pre-trained model to a target task by assessing the impact of the pre-trained model's representations on a reference task. By "transforming" the reference task into the target task, the paper produces a bound on the training loss of the target task

**Strengths:**

1. The paper presents both theoretical and empirical analysis of a problem
2. Some of the provided experiments are reasonably comprehensive
3. The idea of tackling the transferability problem by introducing a reference task seems novel AFAIK

**Weaknesses:**

My main issues with this paper are as follows.
1.  **Utility of the method** :  I was initially excited by the following sentence in the introduction of the paper -- *"Moreover, unlike previous SbTE metrics, task-relatedness can be estimated even without labeled target data, making it suitable for unsupervised transferability estimation, highlighting the advantage of a reference task as used in our analysis"*. However,  it seems that Algorithm 1 requires access to the target labels $\mathcal{Y}_{T}$. This calls into question this stated advantage over the SbTE methods. Specifically, SbTE methods like TransRate and LEEP do not require  training on a reference task which not only comes with extra overhead but whose selection requires a-priori meta-intuition about which tasks are similar to the target task. Also, none of the empirical evaluations presented actually show that the proposed method is superior to SbTE approaches for choosing models form a zoo. Specifically, the paper presents hard to interpret correlation measures in Table 1 as proof of transferability -- when a superior test would be compare deltas in final training performance of the models the models that each method select from a zoo.
2. **Correctness / Impact of claims**:
  [a]. From line 238 - 241: *However, since finding data semantically related to the target task may not always be possible we choose a reference task with the same number of classes as the target and fix that matrix B to a random permutation of identity (making the label mismatch term zero) and D to the prior of the reference task, learning only the transformation A, in our experiments.*. This seems to undercut the utility of the analysis presented. Why does this work ? Why even discuss these terms if they are effectively fixed ?
  [b]. It feels like the insight of *Highly related reference–target task pairs, based on task-relatedness, achieve higher transferability coinciding with the semantic relatedness between tasks.* is just punting the problem of transferability estimation to task similarity estimation (and now calling on practitioner intuition to decide which tasks are highly related). Thus it does not seem that the authors are actually effectively solving the problem of transferability estimation -- they are just transforming it into a different problem.
  [c].  The authors claim  "tight bounds / small gaps between task related and transferability" but this is based on (imo) just eyeballing the graphs in  Figure 3 and 9. We are not provided with a rigorous way to interpret the gap. Eg -- a very simple way to artificially present a small gap is to just plot the the losses at random init of the classification head for the the reference and target tasks that have the same number of class labels.
3. **Strength of underlying technical assumptions**:
  [a].  the K_r > K_t seems like a strong assumption that limits the methods applicability. What if the most similar task to the target task has K_r < K_t.
  [b]. Also as mentioned above in 238 - 241 it seems the label and class-prior matching component of the algorithm effectively has no influence on the final algo results.


Score updated after discussion

**Questions:**

1. Work focuses  on classification based tasks -- could you discuss how this applies to generative tasks
2. Can you discuss how faithful the estimates / method is when K_r < K_t  ?
3. The analysis is for training error -- how is transferability estimation affected if there is overfitting to the reference task ?

**Limitations:**

N/A -- paper presents some limitations

---

> ### Author Rebuttal · Authors · 2024-08-06
>
> Thank you for your thoughtful review. We respond to your concerns below:
>
> > Utility: A) I was ... methods. B) Specifically, ... target task.
>
> A: While Alg. 1 requires labels of the target task, we discuss in Lines 329-347 how to use Alg. 1 for the case when target labels are unavailable. The main idea is to use the transformation model to obtain the pseudo-labels (Line 337) of the target data in the label space of the **target** task. (Note that this is different from the pseudo-labels of the target data in the label space of the **source** task as used by LEEP [34]). Using these pseudo-labels of the target data as a proxy for $Y_T$, we can now use Alg. 1. Since no other SbTE method (to the best of our knowledge) can work without target labels, we stated this in the introduction of the paper. We will update the input of Alg. 1 appropriately to clarify this.
>
> B: We emphasize that the reference task does not need to be similar to the target task to use our methodology for estimating transferability. Any benchmark/publicly available dataset can serve as the reference dataset and its use is not a bottleneck for our analysis/approach. Instead, it makes it possible to study transferability provably, enables us to estimate transferability without target labels, and provides insights into improving the transferability of a model to downstream tasks.
>
> > None of ..
>
> While the best way to evaluate transferability is to fine-tune a model end-to-end on the target task, it is computationally prohibitive. E.g., You et al., 2021 [54] show that end-to-end fine-tuning of a ResNet-50 model on the Aircraft dataset, with hyperparameter search, requires about a day’s worth of computation (Sec 5.5 and Table 5 of You et al., 2021). This is the motivation for most  SbTE approaches that efficiently estimate transferability without end-to-end fine-tuning.
>
> Previous works [5, 34, 25, 54, 49] reported the Pearson correlation coefficient between the accuracy of models after end-to-end fine-tuning and the proposed scores. A high correlation of an SbTE score indicates that comparing the scores of various models and choosing the one with the highest score is most likely the best-performing model on a given target task. Thus, we report correlation coefficients. (Since task-relatedness is based on loss, a higher negative correlation is desirable for us.)
>
>
> > [a]. From line 238 - 241..
>
> We ablate the use of different transformations and measure the effect of learning each of them in Appendix C.1. Our experiments show that minimizing the conditional entropy by learning transformation B prefers a sparse mapping between the classes of the reference and the target tasks. As a result, one class from the target is mapped uniquely to one class from the reference task, and the priors of extra reference classes are reduced to zero (This is illustrated in Fig. 7 of Appendix C.1.1). Based on this, we use a reference task with the same number of classes as the target task and fix the matrix B to be a random permutation of the identity matrix.
>
> Thus, while each term in the bound has its impact, fixing the matrix B and choosing a reference task with the same number of classes as the target, we simplified the optimization problem in Eq. 1 and made it efficient for practical usage.
>
> > [b]. It feels..
>
> The main purpose of the experiments in Sec. 4.2 is to demonstrate the validity of our analysis and the fact when the target task is a transformation of the reference task, a model’s transferability to the target task can be provably explained based on that of the reference task. For e.g., for an encoder trained on MNIST, ground truth transferability to USPS is better than that to FMNIST (Fig. 4(a) right). This is also what is suggested by our task-relatedness analysis/metric using MNIST as the reference task (Fig. 4(a) left). Thus, as mentioned in Fig 1, given a pre-trained encoder and a reference task, we can provably explain the encoder’s transferability to those target tasks that are transformations of the reference task.
>
> For the practical purposes of estimating transferability on the other hand, any reference task can be used (i.e., any publicly available or benchmark data) to compute task-relatedness. As shown in Sec 4.3, the use of a reference task to estimate transferability leads to a more stable estimate of transferability regardless of the number of target samples used as well as enables estimating transferability without access to target labels. Thus, while being theoretically sound, task-relatedness provides practical advantages over previous transferability estimation methods.
>
> > [c] The authors..
>
> A small gap in transferability implies that the difference between the left-hand side of the bound in Theorem 3 and the right-hand side computed by learning the transformations via Alg. 1 is small. While we agree that losses at random initialization for reference and target loss might show a small gap, it will not be very useful. Thus, we measure the left-hand side of the bound after training on the target task and then compare task-relatedness with this value. The target loss presented as the blue bars in Figs 3 and 9 show this value.
>
>
> > The $K_r > K_t$ ... when $K_r < K_t$ ?
>
> Please see our joint response.
>
> >  classification based ... generative tasks
>
> Please see our joint response.
>
> > The analysis..
>
> This seems to be a misunderstanding since our analysis is for expected loss. The classifiers $h_R$ and $h_T$ are learned on the training data and we report transferability and task-relatedness values on the test data from the two tasks similar to any ML pipeline.
>
> When there is overfitting on the reference task, we can expect the reweighted reference task loss (computed on test data) to be higher, which may lead to a larger gap between the actual and predicted transferability. We emphasize that this is not unique to our method as any analysis based on a source/reference task (e.g., [6,7]) will behave similarly.

---

> > ### Comment · Reviewer_3SCA · 2024-08-12
> > **Response**
> >
> > Thanks for your detailed response to my review.  Please see my responses below
> >
> > > `A: While Alg. 1 requires labels of the target task, we discuss in Lines 329-347 how to use Alg.`
> >
> > Hmm -- this is very confusing / seems contradictory. First, note that Algo 1, uses a random permutation matrix as B.
> > Next, in lines in Lines 329-347 requires you to use B to compute the pseudo-labels. But 156 - 166 requires using target labels to specify B. So are you saying that to compute pseudo labels you use a random B ?
> > Please clarify if that is not the case, and if that is the case, what are the error bars on the results in Table 2 ? I would imagine some non-trivial sensitivity to the choice of B.
> >
> > > `B: .. We emphasize that the reference task does not need to be similar to the target task to use our methodology for estimating transferability. Any benchmark/publicly available dataset can serve as the reference dataset and its use is not a bottleneck for our analysis/approach.`
> >
> > This is also confusing to me -- and seems a bit "free-lunchy".  Theorem 3 on line 213 clearly gives **an upper bound** that depends on the label and distribution.  I would imagine that the choice of the reference tasks affects the looseness of this upper bound -- we can simply think of the reference task as a variable that we are minimizing the upper bound over to make it tighter.
> > Thus, the quality of your transferability estimate does depend on the *appropriate* choice of reference task.
> > Please clarify if I am mistaken.
> >
> > > `While the best way to evaluate transferability is to fine-tune a model end-to-end on the target task, it is computationally prohibitive.`
> >
> > I would like to respectfully push back on this. You could have set up much smaller scale experiments to validate that the transferability scores are actually good for a reasonably sized model selection problem.
> >
> > > `We ablate the use of different transformations and measure the effect of learning each of them in Appendix C.1. Our experiments show that minimizing the conditional entropy by learning transformation B prefers a sparse mapping between the classes of the reference and the target tasks. .... Based on this, we use a reference task with the same number of classes as the target task and fix the matrix B to be a random permutation of the identity matrix.`
> >
> > As you mention in Appendix C.1 this is done on a toy problem -- it's a big leap to assume that this is the right thing to do for a more realistic problem without a bit of preliminary / small scale experimentation. Next, even if there is a sparse 1-1 mapping between classes, it is very strange to me that you decided that the exact mapping is not important and rather a random one would suffice. Basically, I understand your restriction to the set of `permutation matrices` but not your assumption that any random one of these matrices would suffice

---

> > > ### Author Response · Authors · 2024-08-13
> > > **Invitation for further discussion**
> > >
> > > Dear reviewer,
> > >
> > > We sincerely appreciate your time and effort in reviewing our manuscript. Through our rebuttal we clarified your concern around making Alg. 1 suitable for transferability estimation when target labels are unavailable. We also justified the reason for reporting the correlation coefficient for the problem of transferability estimation in line with all prior SbTE works [5, 34, 25, 54, 49]. Lastly, our latest response also provided clarification and more empirical evidence showing that fixing some of the transformations (such as fixing the matrix B to a random permutation) eases the combinatorial nature of the optimization problem in Eq. 3 without significantly affecting the value of the transferability estimate.
> > >
> > > As the discussion period is about to end, it would be appreciated if you could let us know of any other concerns we can address at this point. If you believe our responses sufficiently addressed your concerns we would humbly request you to re-evaluate our paper in the light of our discussion and update your rating of our work.
> > >
> > > Thank you immensely for your contributions and thoughtful consideration.
> > >
> > > Regards
> > > Authors

---

> > > > ### Comment · Reviewer_3SCA · 2024-08-14
> > > > **Response**
> > > >
> > > > Thanks for your clarifications
> > > >
> > > > >`For computing the results in Table 2, the matrix B is fixed to a permutation matrix (similar to our other experiments). Thus, there is no variability due to the matrix B.`
> > > > I think the target label free case is clearer now, but I do still think it's important to evaluate the sensitivity of the final result to the initial (random) value of B.
> > > > I think I see the value of the method at least in the case of no target labels present.
> > > >
> > > > >`to use for computing task relatedness, a practitioner can estimate the upper bound using a few benchmark/publicly available datasets and select the reference task as the one that minimizes the upper bound, similar to the approach suggested by the reviewer of treating the reference task as a variable.`
> > > >
> > > > As I mentioned in my initial review, this seems like punting the problem to a new problem of reference task selection and requiring practitioner intuition in selecting the reference task.
> > > > And then there is the case of the computational overhead (if you have to run the algo on several tasks ) of this approach compared to other methods that do not require reference task
> > > >
> > > >
> > > > My questions have been sufficiently clarified by the authors. I will raise my score from 3 --> 6.

---

> > > > > ### Author Response · Authors · 2024-08-14
> > > > > **Response by Authors**
> > > > >
> > > > > We sincerely appreciate the reviewer for engaging in the discussion and are glad that our responses clarified their concerns. We are encouraged that they find practical value in the ability of our method to estimate transferability without target labels.
> > > > > Following the suggestions of the reviewer, we will include the discussion and experiments for the initialization of transformations, presented here, in the paper to further improve the quality of our work.

---

> ### Author Response · Authors · 2024-08-12
> **Response by Authors**
>
> We thank the reviewer for their response to the rebuttal.
>
> > Are ... of B.
>
> The reviewer’s intuition is correct that the pseudo labels are computed using the $B$ matrix provided at the initialization of Alg. 1 for the first time. But as Alg. 1 progresses the matrix $B$ will also be updated. As a result, the pseudo labels of the target data are also updated. This can be thought of as an additional step in Alg 1 as follows:
>
> Step 2a: If target labels ($y_T^j$) are not available then compute $y_T^j = arg max_{y \in Y_T} Bh_R(A^{-1}x_T)$ for all $j = 1, …, n_T$.
>
> For computing the results in Table 2, the matrix B is fixed to a permutation matrix (similar to our other experiments). Thus, there is no variability due to the matrix B.
>
> > I would ... mistaken.
>
> The reviewer's intuition is correct. The choice of reference task affects the upper bound as we demonstrated, in detail, in Sec. 4.2. Specifically, when the reference and target tasks are more related (in terms of our task-relatedness metric) the upper bound is smaller.
>
> However, finding the most related reference task for every target task may not be possible in practice. Since our analysis does not impose any restriction on the choice of the reference task to use for computing task relatedness, a practitioner can estimate the upper bound using a few benchmark/publicly available datasets and select the reference task as the one that minimizes the upper bound, similar to the approach suggested by the reviewer of treating the reference task as a variable.
>
> > I would like ... problem.
>
> This seems to be a misunderstanding. SbTE methods as per [54] produce a score for each model (ideally without end-to-end fine-tuning on the target) that correlates well with end-to-end accuracy. This allows selecting the top-performing model by simply evaluating these scores. Thus, all prior works report correlation coefficients similar to our results in Table 1.
>
> Moreover, scores produced by SbTE methods do **not** numerically approximate the accuracy after end-to-end fine-tuning. For example, the results in Fig. 4 of [54] show that the value of SbTE metrics (y-axis of the plot) such as LEEP [34], NCE [50], and LogMe [54] lies in the range [-0.8,-0.3], [-0.45, -0.25], and [0.935, 0.95] respectively for the Aircraft dataset whereas the end-to-end fine-tuning accuracy (x-axis of the plot) lies in the range ~[72.5, 87.5] for the various models.
>
> It would be great if the reviewer could clarify their question if we have misunderstood them.
>
> > It is very ... suffice.
>
> The experiment in Appendix C.1 is a small-scale experiment as it is in the same setting as in Sec 4.1 with the ResNet-18 model. It is intended to be an ablation study to understand how different transformations affect the upper bound.
>
> The label mismatch term of Theorem 3 is minimized with a sparse B matrix corresponding to a one-to-one mapping between the classes of the reference and target tasks. However, due to the combinatorial nature of the problem finding the exact permutation is infeasible. Regardless of that, the linear transformation $A$ can align the distributions of the representation of the target to any permutation of the reference classes quite accurately. This is primarily due to the high dimensionality of the representation space where the linear transformation is applied. As a result, the value of the bound differs only by little for the different permutations of the reference task’s classes. This is indeed a very interesting outcome that makes our bound usable without requiring the exact semantic matching between the classes of the reference and the target classes.
>
> Below, we present an empirical evaluation to show that the difference in the bound computed with true vs random permutation is small. We use MNIST as the reference task and USPS as the target task (and vice-versa). We compare our results in a setting where only $A$ is learned and $B$ is set to an identity matrix and when $B$ is set to a random permutation matrix. Note that the identity matrix corresponds to the correct mapping between the classes of MNIST and USPS tasks (both contain digits from 0 to 9).
>
> We find that the upper bound obtained when $B$ is fixed to identity is only marginally better than the case when $B$ is a random permutation. Specifically, the difference between the bound when $B$ is fixed to a random permutation and when $B$ is an identity matrix is 0.10 for the MNIST→USPS task and 0.17 for the USPS→MNIST task. The primary reason for the decrease in the upper bound comes from the reduced distribution mismatch term.
>
> While the upper bound improves slightly when the ideal matching between the labels is known, such a mapping may not be known when the labels of the tasks are not related, e.g., for FMNIST and MNIST. Thus, fixing $B$ to a random permutation matrix yields a reliable estimate of transferability in most cases.
>
>
> We would be happy to address any further concerns the reviewer may have.

---

### Author Rebuttal · Authors · 2024-08-05

We thank all the reviewers for their valuable feedback and insightful questions. We are encouraged that all the reviewers found the main contribution of the paper of rigorously analyzing transferability to target tasks in a cross-domain cross-task transfer learning setting using task-relatedness novel (3SCA), intuitive (E8p5), practically valuable (sD5Z); and our empirical evaluation comprehensive (3SCA, sD5Z).

We are glad that reviewers E8p5 and dEjE appreciated the significance and contributions of our work and recommended acceptance. While 3SCA and sD5Z recommended reject and borderline reject, respectively, the weaknesses pointed out by them are primarily clarification questions about the paper’s setting and utility, which we have addressed comprehensively in this rebuttal. Please see our comments below for answers to each reviewer's specific questions. Along with these, we have included additional results comparing our method with OTCE and Hscore [a] for the end-to-end transferability estimation problem considered in Sec 4.3.

Based on these answers and clarifications we have addressed the concerns of each of the reviewers. Hence, we hope reviewers will consider our answers, increase their ratings, and recommend acceptance. If additional questions arise during the discussion phase, we are just a post away and would be happy to address them.

> **[dEJE, sD5Z]: Additional results for OTCE and Hscore**

OTCE [49] and [a] propose score-based transferability metrics for the problem of end-to-end transferability estimation similar to those discussed in Sec. 4.3. Specifically, as discussed in Sec. 2, OTCE proposes a linear combination of negative conditional entropy between the label sets of the source and target tasks computed using OT-based coupling matrix and the Wasserstein distance between the representations of the two tasks. [a] on the other hand, solves the HGR maximum correlation problem to analyze transfer learning in the setting when the source and target domain have the same features, similar to [50]. They propose to use normalized Hscore as a way to estimate transferability. We have expanded our empirical section to include these baselines in Sec 4.3. We emphasize that both these methods are specifically proposed for score-based transferability estimation and do not provide a rigorous analysis of transferability, unlike our approach.

The comparisons to these were omitted due to the requirement of auxiliary tasks required to obtain hyperparameters crucial for OTCE’s performance as mentioned by the authors and some considered baselines such as LEEP and LogMe had already shown improvements over Hscore. However, we have expanded our empirical results to include these methods.

For OTCE, we follow the official code and compute the recommended OT-based NCE score and OTCE score ($\lambda_1=-0.0001$ and $\lambda_2=-1$) using 4000 randomly selected training samples from the two tasks. For the source task, we subsample data from the same number of classes as the target task. For the H-score, we use the official code to compute the metric. The results for these two new metrics are presented in the attached pdf corresponding to Table 1 of the paper which evaluates the Pearson correlation of the metrics to the end-to-end fine-tuning accuracy of five models and Fig 5 of the paper which evaluates the sensitivity of the metrics to different numbers of target samples.

Consistent with the results in Table 1 and Fig 5 of the paper, task-relatedness achieves a high correlation to end-to-end fine-tuning accuracy better than most previous SbTE methods. This correlation also remains stable regardless of the number of target samples, unlike most other SbTE methods whose correlation degrades severely in scenarios with fewer samples from the target task.

> **[3SCA, sD5Z]: Does the proposed method apply to the settings where the target task has more classes than the reference task?**

Yes, since this assumption does not affect the analysis. We assumed it since we focus on explaining transferability with the performance/classifier of the reference task. Thus, scenarios where the reference task has fewer classes than the target would intuitively lead to a poor understanding of transferability. While task-relatedness can be computed in this setting, the presence of fewer classes in the reference task can increase the conditional entropy and the distribution mismatch terms. This is because the mass of the extra target classes will be split across multiple reference task classes leading to a higher distribution mismatch. The conditional entropy will be higher because a sparse mapping (such as one-to-one) between reference and target task classes cannot be obtained.

Practically, since a practitioner can choose any reference task to evaluate task-relatedness, it's preferable to use one with more classes to get a better estimate of transferability.

> **[3SCA, sD5Z]: Does the proposed metric apply to non-classification tasks?**

The popularity of transfer learning in the classification setting is the primary reason we analyze this setting. The transformation model used in our work includes components such as labels or prior changes, which are most suitable for classification tasks. While the distribution mismatch component could be useful beyond classification tasks, it is unclear as to what form of distribution divergence is most suitable. Developing an appropriate transformation model for auto-regressive tasks and the performance metrics associated with them requires further research. Thus, we believe that it is an excellent suggestion by the reviewers to extend the analysis to other non-classification tasks, however, it is non-trivial and currently out of the scope of this paper.

**New References**

[a]: An information-theoretic approach to transferability in task transfer learning.

---

### Public Comment · ~Moein_Sorkhei1 · 2024-12-15
**Public comment**

I congratulate the authors on the acceptance of their paper. The problem of transferability estimation is highly relevant in the field, and I appreciate the effort to propose a new method that advances this research area.

After reading the paper and reviewers’ feedback, I am still left with some questions that I would like the authors to clarify:

**Questions**

**Use of Reference Task:**

The proposed method requires access to a reference task, which could limit its practicality. Many off-the-shelf foundation models are now trained on proprietary datasets that are not publicly accessible.
1. I believe this is a limitation of the proposed method, as the majority of the compared transferability estimation methods do not require access to the source data. What is your insight on this?
2. How does the method perform in scenarios where source datasets are unavailable?
3. Even if the source data is available, self-supervised models are trained without using labels. How does the proposed method handle self-supervised models, which are trained without labels, differently from supervised models? In my opinion, the two major components of the proposed method, that is Reweighted Reference Loss and Label Mismatch become irrelevant for SSL methods. What is your opinion?

__Practical use__

From Table 1 and Figure 5, it appears that LEEP often performs well, even surpassing the proposed method in Table 1. Yet, LEEP offers some advantages: 1) it does not require a reference set, 2) simply accounts for task differences, and 3) avoids optimization.
What practical advantages does the proposed method offer over LEEP to motivate its use for me as a practitioner?


__Target Datasets Without Labels__

The paper mentions using pseudo-labels generated by the transformed classifier as a proxy for target labels. However, off-the-shelf clustering methods could similarly generate pseudo-labels on the target data, enabling other transferability estimation methods to be applied on unlabeled target.
Could the authors clarify how this is a distinct advantage of their method?
Further, how this step can be handled if one is transferring an SSL model, which does not directly produce class labels?






**Suggestions**

__Technical statements__

The claim, "Since end-to-end fine-tuning is costly (takes almost a day to fully fine-tune a single model on a single target task...)," seems overly broad and potentially inaccurate in many scenarios.
Fine-tuning time varies significantly depending on factors such as the size of the model and the target dataset, among others.
What is your insight?

---

> ### Public Comment · ~Akshay_Mehra1 · 2024-12-23
> **Response to the public comment**
>
> Thank you for your interest in our work; we appreciate your questions. We have answered your questions below. Please feel free to reach out if you have any further questions.
>
> > The proposed ... this?
>
> Our work's primary contribution is understanding on which tasks transfer learning can lead to high-performing models. To answer this question, we consider a reference task that achieves high performance after fine-tuning. Through our upper bound in Theorem 3, we show that performance on any target task can be provably related back to this reference task. Crucially, the reference task may or may not be the source task used to train the pre-trained model, and any publicly available dataset can be considered the reference task.  Lastly, various previous transferability estimation methods such as NCE[56] and OTCE [55] have also utilized the source task for computing their metric for transferability.
>
> In our empirical section, we show that task relatedness is also an effective metric for the pre-trained model selection problem. We also emphasize that, unlike our work, an analytical understanding of transferability in the cross-domain cross-task setting is not the focus of other score-based transferability estimation methods.
>
> > How ... unavailable?
>
> As mentioned above, our analysis does not restrict the choice of the reference task to use for computing task relatedness. Any benchmark or publicly available dataset can serve as the reference task.
>
> > Even if ... your opinion?
>
> In Figures 3 and 9, we show experiments with various self-supervised models such as MOCO, SWAV, and MAE. We used Imagenet as the reference task for these experiments. Since the reference task can be any task, all terms in the bound are still meaningful.
>
> > From Table 1 and Figure 5, ... as a practitioner?
>
> We see in Figure 5 that LEEP performs worse in scenarios where there is less target data available. Comparatively, our method produces consistent correlation at different percentages of target data.
>
> While we agree that LEEP indeed offers the practical advantages you mentioned, it is sensitive to the amount of target data available for transferability estimation, and it only provides limited insights into analytically explaining transferability, unlike our approach.
>
> > A) The paper mentions ... of their method? B) Further, how this step can... class labels?
>
> A) While we agree that practically various alternatives such as off-the-shelf clustering methods could be used to obtain pseudo-labels, our task transfer approach does not require using these to handle unsupervised transferability estimation, unlike other score-based methods.
>
> B) For pre-trained SSL models, the predictions from $h_R$, which is the classifier trained on the reference task, can be used to obtain the pseudo-labels using $\arg \max_{y \in Y_T} Bh_R(A^{−1}(z_T))$ as mentioned in Sec. 4.3.
>
> > The claim, ... What is your insight?
>
> While end-to-end fine-tuning depends on various factors, an in-depth analysis from previous works such as Table 4 in LogMe [61] has shown that it is extremely costly to find the best end-to-end fine-tuned model for a single target task along with the best hyperparameter search. This has been the basis of various score-based transferability estimation approaches we cited in our paper.

---

### Decision · Program_Chairs · 2024-09-25

**Decision:**

Accept (poster)

**Comment:**

The paper addresses task-relatedness in cross-domain cross-task transfer learning. The main contribution is an upper bound on transferability of the representations of pre-trained models to downstream tasks measured in terms of their relatedness to a given reference task. The proposed task-relatedness works without requiring target labels, and the notion of a reference task has been well received.

Initial reviews raised several issues, including clarifications of the contributions, limited empirical evidence, and related work comparisons. The rebuttal successfully addressed most of these concerns, leading to two reviewers raising their scores. After careful deliberation, the decision was made to accept this paper - congratulations to the authors! It is crucial, however, that all the promised improvements and clarifications including the related work, sensitivity analysis and presentation clarity, are addressed in the final version of the paper.